# FASTRAIN-GNN: Fast and Accurate Self-Training for Graph Neural Networks

**Amrit Nagarajan**  *nagaraj9@purdue.edu*
*School of Electrical and Computer Engineering*
*Purdue University*

**Anand Raghunathan**  *raghunathan@purdue.edu*
*School of Electrical and Computer Engineering*
*Purdue University*

**Reviewed on OpenReview:** *https://openreview.net/forum?id=1IYJfwJtjQ*

## Abstract

Few-shot learning with Graph Neural Networks (GNNs) is an important challenge in expanding the remarkable success that GNNs have achieved. In the transductive node classification scenario, conventional supervised training methods for GNNs fail when only few labeled nodes are available. Self-training, wherein the GNN is trained in stages by augmenting the training data with a subset of the unlabeled data and the predictions of the GNN on this data (pseudolabels), has emerged as a promising approach to few-shot transductive learning. However, multi-stage self-training significantly increases the computational demands of GNN training. In addition, while the training set evolves considerably across the stages of self-training, the GNN architecture, graph topology and training hyperparameters are kept constant, adversely affecting the accuracy of the resulting model as well as the computational efficiency of training. To address this challenge, we propose FASTRAIN-GNN, a framework for efficient and accurate self-training of GNNs with few labeled nodes. FASTRAIN-GNN performs four main optimizations in each stage of self-training: (1) Sampling-based Pseudolabel Filtering removes nodes whose pseudolabels are likely to be incorrect from the enlarged training set. (2,3) Dynamic Sizing and Dynamic Regularization find the optimal network architecture and amount of training regularization in each stage of self-training, respectively, and (4) Progressive Graph Pruning removes selected edges between nodes in the training set to reduce the impact of over-smoothing. On few-shot node classification tasks using different GNN architectures, FASTRAIN-GNN produces models that are consistently more accurate (by up to 4.4%), while also substantially reducing the self-training time (by up to 2.1×) over the current state-of-the-art methods. Code is available at `https://github.com/amrnag/FASTRAIN-GNN`.

## 1 Introduction

Advances in supervised training of Graph Neural Networks (GNNs) have revolutionized the field of graph-based learning. As a result, GNNs find wide-spread applications in several fields, ranging from friend and product recommendations in social media networks to molecular property prediction for drug discovery. However, real world graphs are often sparsely-labeled, and training GNNs in a few-shot setting remains challenging. While conventional supervised training methods for GNNs (Kipf & Welling, 2017; Velickovic et al., 2018; Hamilton et al., 2017) work well in the presence of large amounts of labeled training data, their performance drops rapidly as training data becomes more scarce due to insufficient propagation of information from labeled data.

Two main classes of semi-supervised training algorithms have been proposed to address this challenge in the context of node classification using GNNs: pre-training and self-training. Pre-training is effective in the inductive node classification setting (where new, unseen graphs are presented at testing time; the goal is to label the unlabeled nodes in the new graphs). Since training GNNs with good generalization performance is the key for effective inductive node classification, large amounts of non-targeted labeled data can be used to pre-train the GNN, followed by a few iterations of fine-tuning on the limited task-specific data. As a result, prior works on pre-training GNNs (Hu et al., 2020; Lu et al., 2021) have demonstrated significant improvements on few-shot inductive node classification. On the other hand, the transductive node classification setting (a small number of nodes in a large graph are labeled; the goal is to label the unlabeled nodes in the graph) benefits more from self-training. In particular, since the entire graph (consisting of both labeled and unlabeled nodes) is available at training time, effective transductive node classification is achieved by targeted training on the input graph (learning from the topology and vast amounts of unlabeled data in the graph). Consequently, prior works on self-training (Li et al., 2018; Sun et al.) have demonstrated good performance on few-shot transductive node classification.

In this work, we focus on self-training GNNs for few-shot transductive node classification, which is outlined in Fig. 1. Self-training is performed in multiple stages, where a GNN model is trained in each stage using the current training set. At the end of each stage, the training set is enlarged by adding nodes that are likely to be predicted correctly by the trained GNN along with their associated pseudolabels. Typically, confidence is used as the metric to assess correctness of predictions — if the confidence of the GNN in predicting a node is above a threshold ($T_c$), the node is added to the training set for the next stage. The enlarged training set results in a greater fraction of nodes in the graph participating in the training process (through message passing), leading to better node representations, and hence, better classification accuracy. While the conventional self-training framework has led to substantial accuracy improvements over single-stage supervised training (Li et al. (2018); Sun et al.), it presents significant computational overheads during training. For example, on {Cora, Citeseer and Pubmed} datasets with 4 labeled nodes/class, 4-stage self-training takes {$4.6\times$, $5.2\times$ and $5.4\times$} longer than single-stage fully-supervised training (not exactly $4\times$ longer, since the training set is also enlarged in each stage, see Appendix A). In addition, while prior efforts have led to significant improvements in the accuracy of pseudolabels added to the training set in each stage, the GNN architecture, graph topology and training hyperparameters (such as strength of regularization) have been kept constant throughout self-training, adversely impacting both the accuracy of the model and the efficiency of the self-training process.

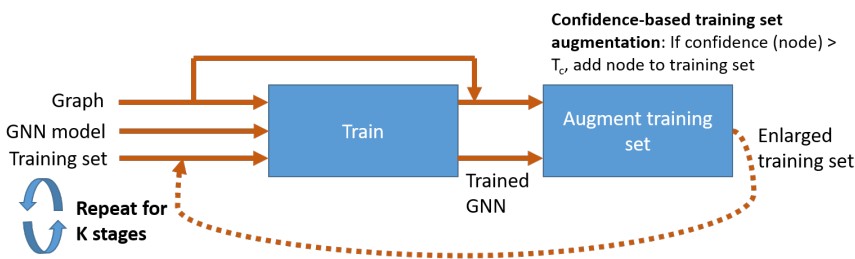

Figure 1: **An overview of self-training for GNNs.**

We present FASTRAIN-GNN, a framework for fast and accurate self-training of GNNs. FASTRAIN-GNN adds four main optimizations to the conventional self-training loop. First, we observe that incorrect training labels have a large detrimental effect on the accuracy of the trained model, and propose Sampling-based Pseudolabel Filtering (SPF) to improve the accuracy of pseudolabels added to the training set. We also propose Dynamic Regularization (DR) to reduce the impact of residual errors in training labels (after SPF) during self-training. Second, we note that the "layer effect" in GNNs (Sun et al.) plays a major role during the different stages of self-training. Deep GNNs outperform shallow GNNs when training on few labeled nodes due to larger message passing coverage (leading to better propagation of information from labeled nodes) in deep GNNs. However, as the number of labeled nodes increases, the larger message passing coverage leads to oversmoothing in deep GNNs, where nodes belonging to different classes acquire similar representations and become less discernible from each other. As a result, shallow GNNs outperform deep GNNs when training on large numbers of labeled nodes. Based on this observation, we propose Dynamic Sizing (DS) to adapt the network architecture across the stages of self-training. Finally, we observe that inter-class edges in the graph

add noise during message passing, and propose Progressive Graph Pruning (PGP) to incrementally prune inter-class edges between nodes in the training set. While different pruning techniques such as the Lottery Ticket Hypothesis (Chen et al., 2021) and sampling (Zhang et al., 2023) have been proposed for improving the training and inference efficiency of GNNs, SPF and PGP use pruning for achieving gains beyond just efficiency. SPF randomly prunes a small subset of edges in the graph to test the robustness of predictions to perturbations in the graph structure, thereby filtering out incorrect training labels. On the other hand, PGP prunes inter-class edges to reduce over-smoothing from noisy message passing, thereby leading to better generalization performance. In addition to producing more accurate self-trained models, the four proposed optimizations also considerably improve the efficiency of the self-training process. SPF reduces the size of the training set, while DS, DR and PGP result in smaller GNNs being trained on sparser graphs.

We summarize our main contributions as follows:

- We introduce FASTRAIN-GNN, a framework for fast and accurate self-training of GNNs. In particular, FASTRAIN-GNN focuses on optimizing the GNN architecture, training data, training parameters, and the graph topology during self-training.

- We propose Sampling-based Pseudolabel Filtering to generate smaller training sets with more accurate pseudolabels in each stage.

- We propose Dynamic Sizing and Dynamic Regularization to overcome the layer effect in GNNs and reduce the impact of incorrect training labels, respectively.

- We propose Progressive Graph Pruning to improve the information-to-noise ratio during message passing and reduce the impact of over-smoothing in GNNs.

- Across four different transductive node classification datasets using different GNN architectures and with varying numbers of labeled nodes, we demonstrate that FASTRAIN-GNN simultaneously improves both the accuracy of the self-trained models and the efficiency of self-training over the previous state-of-the-art.

## 2 Related Work

Prior work on self-training GNNs has almost exclusively focused on the problem of generating more accurate training labels (pseudolables) in each stage of self-training. Li et al. (2018) proposed four different techniques for generating more accurate training labels. Co-training trains a random walk model along with the GNN to explore the graph structure, and adds the most confident predictions of the random walk model to the training set. Self-training adds the most confident predictions from the GNN to the training set. Union adds a union of the most confident nodes found by the GNN and the random walk model, while Intersection adds the common subset of most confident predictions of both models to the training set. Recent works have proposed improvements to these original baselines. Some notable examples include DSGCN (Zhou et al., 2019), which uses negative sampling and confidence-based weighting of samples to reduce the impact of incorrect training labels. M3S (Sun et al.) introduces a checker based on DeepCluster (Caron et al., 2018) that verifies the correctness of the most confident predictions before adding them to the training set. IFC-GCN (Hu et al., 2021) rectifies incorrect pseudolabels based on feature clustering. GraphMix (Verma et al.) trains a fully-connected neural network along with the GNN to improve the accuracy of GNN predictions. CGCN (Hui et al., 2020) validates pseudolabels based on predictions from a graph clustering network that combines variational graph auto-encoders with Gaussian mixture models. Self-enhanced GNN (Yang et al., 2021) enlarges the training set using the most confident predictions from an ensemble of GNNs. Finally, the current state-of-the-art method (see Appendix C) for self-training is CaGCN (Wang et al., 2021), which uses confidence calibration (Guo et al., 2017) to improve the accuracy of the most confident predictions. Complementary to these works, we focus on the problem of how the enlarged training set can be used to efficiently self-train GNNs to high accuracies, with a specific focus on optimizing the GNN architecture, training parameters and graph topology during self-training. In addition, we also propose a simple filtering method based on robustness to sampling that quickly identifies and removes nodes that are likely to be incorrectly labeled from the enlarged training set. While sampling has been demonstrated to be an effective regularizer for improving both the training efficiency (Chen et al., 2018; Ramezani et al., 2020; Zhang et al.,

2023) and generalization ability (Hamilton et al., 2017; Li et al., 2022) of GNNs, we show that sampling can also be used to test the robustness of GNN predictions to small perturbations in the graph structure.

## 3 The **FAST**RAIN-GNN framework

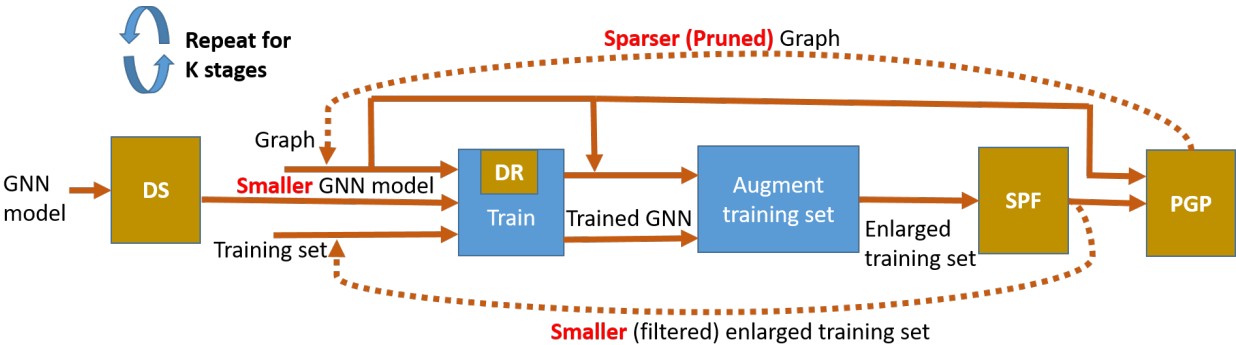

Figure 2: **An overview of the FASTRAIN-GNN framework.**

We present FASTRAIN-GNN, a framework for efficient and accurate self-supervised training of GNNs. Fig 2 presents an overview of the FASTRAIN-GNN framework. First, the appropriate GNN architecture is chosen based on the current size of the training set (DS), and the model weights are randomly initialized. This random initialization is different from the majority of previous works (such as Zhou et al. (2019); Li et al. (2018) etc.) that initialize the model using the trained weights from the previous stage (see Appendix B). Then, the GNN model is trained on the current training set, and the amount of regularization during training is chosen based on the stage of self-training (DR). The training set is subsequently enlarged by predicting the unlabeled nodes with the trained GNN, and adding nodes that are likely to be predicted correctly to the training set. Here, we use confidence-based pseudolabeling, where nodes that are predicted with confidence above a threshold ($T_c$) are added to the training set along with their pseudolabels. The enlarged training set is then filtered based on the robustness of predictions to small perturbations in the graph topology (SPF), to produce a smaller training set with more accurate labels. Finally, inter-class edges between nodes in the training set are progressively pruned (PGP) to improve information-to-noise ratio during message passing (when the pruned graph is used to train the GNN model in the next stage). This process is repeated $K$ times, where each iteration of this loop constitutes one stage of self-training.

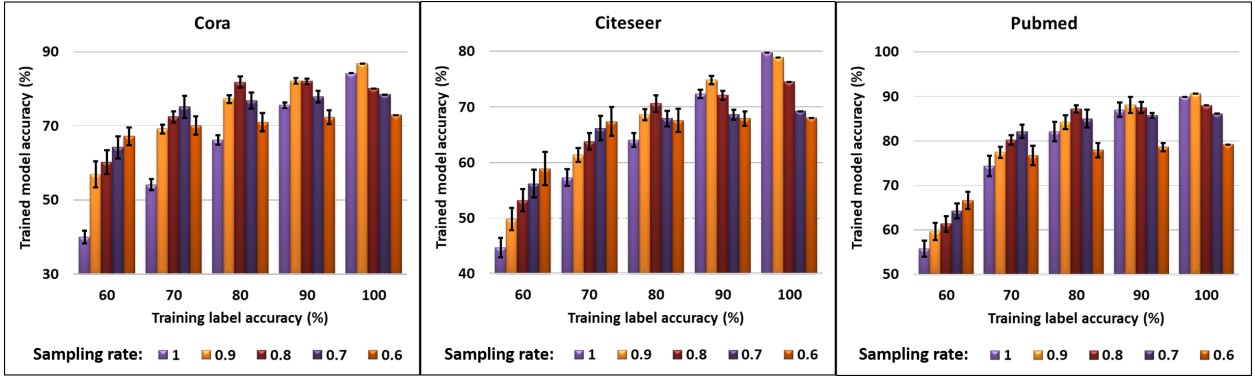

Figure 3: **Impact of incorrect training labels on GCN training.** Results are reported based on fully-supervised training of a 2-layer GCN model, following the settings described in (Rong et al., 2020). A random subset of training labels are changed to an incorrect (random) class for this study, and results are averaged across 100 runs (error bars indicate standard deviation).

In summary, FASTRAIN-GNN adds four main optimizations to the conventional self-training loop – SPF, DS, DR and PGP. We describe these optimizations in the following subsections, with more detailed explanations in Appendix D.

### 3.1 Sampling-based Pseudolabel Filtering (SPF)

SPF is performed immediately after the training set is enlarged by adding nodes with highly confident predictions and their pseudolabels. Since the accuracy of the self-trained model degrades rapidly with decrease in accuracy of the training labels (Fig. 3), it is vital to ensure that the pseudolabels are highly accurate. While confidence is the current state-of-the-art metric for quantifying the correctness of predictions, we observe that the labels generated by confidence-based pseudolabeling still contain substantial noise that has a detrimental impact on the final model accuracy. SPF takes advantage of the fact that correct predictions are significantly more robust to small perturbations in the graph topology compared to incorrect predictions (Fig. 4) to filter out nodes that are likely to have incorrect labels from the training set. Algorithm 1 describes the procedure used to perform SPF. We randomly sample the input graphs with a sampling rate of $spf\_sampling\_rate$ (denoting the fraction of edges that are retained). This is repeated across $spf\_iterations$, with each iteration testing on a different sampled graph. Only those nodes whose predictions are invariant under sampling are included in the filtered training set. As a result, SPF produces a smaller training set with more accurate training labels, leading to improvements in both the efficiency of self-training and the accuracy of the self-trained model.

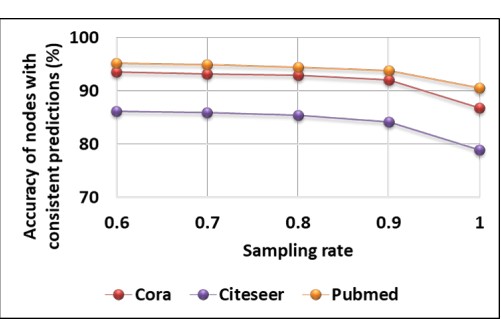

Figure 4: **Accuracy of consistent predictions with different sampling rates.** Results are reported based on full-supervised training of a 2-layer GCN model. Here, "consistent predictions" indicates that node predictions do not change when predicted from 10 different randomly sampled graphs.

---

**Algorithm 1:** Sampling-based Pseudolabel Filtering (SPF)

**Input** : Enlarged training set after confidence-based pseudolabel generation ($T_{init}$), graph ($G$)
**Output:** Filtered training set ($T_{filtered}$)
**1** $spf\_sampling\_rate$ : sampling rate to test prediction consistency
**2** $spf\_iterations$: number of sampling iterations to test prediction consistency
**3** $T_{filtered} = T_{init}$
**4** Initial_predictions = predict(nodes($G$))
**5** **for** *sampling iteration in range(spf_iterations)* **do**
**6**     $G_{sampled}$ = Sample($G$, $spf\_sampling\_rate$)
**7**     Sampled_predictions = predict(nodes($G_{sampled}$))
**8**     **for** *each node in $T_{filtered}$* **do**
**9**         **if** *Sampled_predictions[node] != Initial_predictions[node]* **then**
**10**             $T_{filtered} = T_{filtered}$ - node
**11** return $T_{filtered}$

---

### 3.2 Dynamic Regularization (DR)

While SPF significantly improves the accuracy of training pseudolabels, incorrect labels are inevitable when they are generated from the predictions of a neural network. DR aims to reduce the impact of residual errors in pseudolabels, since even a small number of incorrect labels can lead to large deterioration in accuracy of the trained model (Fig. 3). While overfitting to incorrect labels is problematic for all classes of deep neural networks, we find that it is especially true for GNNs due to the effects of error propagation (Nagarajan et al., 2022). Nodes that acquire representations that cause them to be incorrectly classified during training adversely impact the classification of all their neighbors due to message passing. Therefore, it is vital to

---

**Algorithm 2:** Dynamic Regularization (DR) and Dynamic Sizing (DS)

---

**Input** : Stage of self-training (*stage*), Training set (*T*)
**Output:** GNN model (*gnn*) and sampling rate (*sampling_rate*) to be used in *stage*

**1** *sampling_rate_init* : sampling rate used in the first stage of self-training when only nodes with golden labels are used for training

**2** *sampling_rate_final* : sampling rate used in the later stage of self-training when nodes with both golden- and pseudo-labels are used for training

**3** *deep_gnn* : architecture of the deep GNN used in the initial stages of self-training when only few nodes are used for training

**4** *shallow_gnn* : architecture of the shallow GNN used in the later stages of self-training when large numbers of nodes are used for training

**5** *DS_Threshold* : the size of the training set at which the *deep_gnn* is to be replaced by the *shallow_gnn*

**6 if** *stage == 1* **then**

**7** | *sampling_rate = sampling_rate_init*

**8 else**

**9** | *sampling_rate = sampling_rate_final*

**10 if** *cardinality(T) < DS_Threshold* **then**

**11** | *gnn = deep_gnn*

**12 else**

**13** | *gnn = deep_gnn*

**14** return *gnn, sampling_rate*

---

incorporate techniques that minimize the impact of incorrect labels in the self-training loop to prevent error propagation. Increased regularization has been shown to improve training in the presence of noisy labels (Liu et al., 2020), and we find that sampling (Hamilton et al., 2017; Chen et al., 2018) is an effective regularizer that prevents overfitting to incorrect labels (Fig. 3). We propose DR to dynamically alter the sampling rate during self-training to account for noisy labels in the enlarged training set. The procedure for performing DR is depicted in Algorithm 2. In the first stage of self-training, training labels are golden (100% accurate), and hence, we use high sampling rates (most edges are retained in the graph). We then dynamically reduce the sampling rate as pseudolabeled nodes are added to the training set to account for the reduced accuracy of training labels. In effect, DR provides the dual benefit of resilience to noisy pseudolabels during self-training, and improved efficiency from training on sparser graphs in the later stages of self-training.

### 3.3 Dynamic Sizing (DS)

DS overcomes the layer effect (Sun et al.) in GNNs during the different stages of self-training. Deep GNN models perform better than shallow GNN models when only a small number of labeled nodes are available

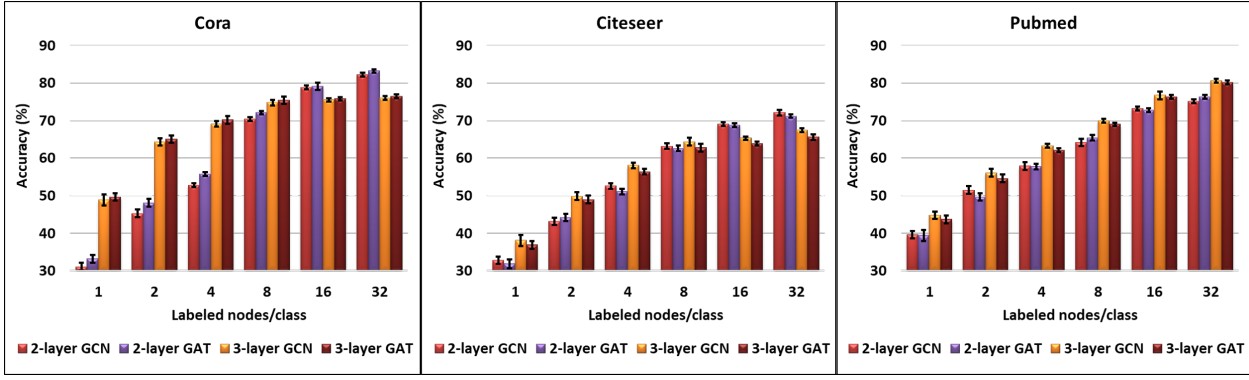

Figure 5: **Demonstration of the layer-effect in GNNs.** Results are averaged across 100 random train-test splits, and error bars indicate the standard deviation.

due to better propagation of label information. In particular, since all L-hop neighbors participate in the prediction of a node through message passing in a L-layer GNN, more nodes participate in the training process in deeper GNNs, leading to better node representations. As the number of training nodes increases, there is greater supervision during training, and hence, shallow GNN architectures start performing better due to greater message passing coverage (Fig. 5). Eventually, under sufficient supervision, shallow GNNs start outperforming deeper GNNs due to the oversmoothing problem in deeper GNNs (Chen et al., 2020a). Due to the smoothing effect of GNN layers, the representations of nodes belonging to different classes become less distinguishable from each other when a large number of labeled nodes are used to train deep GNNs. During self-training, as the training set is progressively enlarged, we observe that the layer effect plays a major role. Initially, when the training set is small, deeper GNNs are more accurate. As the training set is enlarged by generating more and more pseudolabels, shallow GNNs perform better. DS varies the model architecture used in each stage of self-training based on the size of the training set. The procedure for performing DS is also depicted in Algorithm 2. When the size of the training set exceeds a threshold ($DS\_Threshold$), a shallow GNN is used for all further stages.

### 3.4 Progressive Graph Pruning (PGP)

Pruning edges from graphs used for transductive node classification can have a significant impact on training (and inference) efficiency of GNNs due to the exponential complexity of message passing (Chen et al., 2021; Nagarajan et al., 2022). In addition, it has been proven that pruning inter-class edges in homophilous graphs (where connected nodes are likely to have similar labels) improves the information-to-noise ratio during message passing and helps alleviate the over-smoothing problem to an extent (Chen et al., 2020a). In particular, message passing between nodes of different classes causes all nodes in the graph to acquire similar representations (making them indistinguishable). When inter-class edges are pruned, nodes belonging to different classes acquire distinct representations, thereby leading to better class separation, and hence, better classification performance (Fig. 6). When only few labeled nodes are available for training, identifying inter-class edges is challenging. Hence, we propose progressively pruning inter-class edges in each stage based on the pseudolabels of nodes added to the training set. Since nodes added to the training set are significantly more likely to be predicted correctly com-

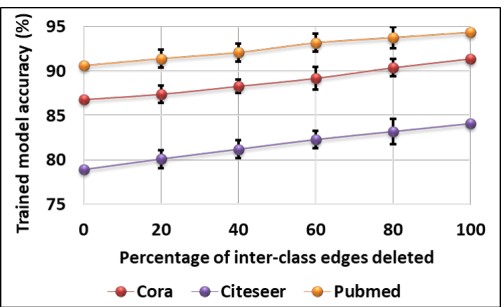

Figure 6: **Impact of pruning inter-class edges between training nodes on accuracy.** Results are reported based on full-supervised training of a 2-layer GCN model. A random subset of inter-class edges are pruned, and results are averaged across 10 runs (error bars indicate standard deviation).

pared to other nodes, restricting PGP to operate only on nodes in the training set minimizes the risk of accidentally pruning intra-class edges.

---

**Algorithm 3:** Progressive Graph Pruning (PGP)

**Input** : Enlarged training set after SPF ($T_{filtered}$), graph ($G$)
**Output:** Pruned graph $G_{pruned}$
1   $Predictions$ = predict(nodes($G$))
2   $G_{pruned} = G$
3   **for** *each node in $T_{filtered}$* **do**
4      **for** *each neighbor of node* **do**
5          **if** *neighbor in $T_{filtered}$* **then**
6              **if** *Predictions[node] != Predictions[neighbor]* **then**
7                  $G_{pruned} = G_{pruned}$ - edge(node, neighbor)

8   return $G_{pruned}$

---

Algorithm 3 demonstrates PGP. After SPF generates an enlarged training set with pseudolabels, PGP then prunes inter-class edges between nodes in the training set. Since the training set is progressively enlarged in each stage of self-training, PGP progressively prunes inter-class edges during self-training, thereby making each iteration of the self-training loop faster than the previous iteration. When GNNs are self-trained on heterophilous graphs (where connected nodes are likely to have dissimilar labels), we modify PGP to prune intra-class edges instead of inter class edges (see Appendix F).

## 4 Experiments and Results

We implement FASTRAIN-GNN using DGL in PyTorch, and evaluate it on a GeForce RTX 2080 Ti GPU with 11GB memory. We randomly select (labels/class) nodes of each class as training nodes, and report results on the rest of the nodes in the graph (we do not require a separate held-out validation set for any of the FASTRAIN-GNN optimizations). We repeat

Table 1: **Dataset characteristics.**

| Dataset | Classes | Nodes | Edges | Feature size |
|---|---|---|---|---|
| Cora | 7 | 2708 | 5429 | 1433 |
| Citeseer | 6 | 3327 | 4732 | 3703 |
| Pubmed | 3 | 19717 | 44338 | 500 |
| CoraFull | 70 | 19793 | 126842 | 8710 |

this process 100 times for each value of (labels/class), and all results reported in this section are averaged across 100 different training splits (with error bars indicating accuracy range), unless otherwise specified. The datasets used for testing are summarized in Table 1. Since some classes contain too few nodes to get a meaningful train-test split in CoraFull, we do not present results with 16 labels/class. The details of the hyperparameters used in all experiments are presented in Appendix D.

Table 2: **Results of training GCN with different label rates.** The most accurate model without confidence calibration is underlined, while the most accurate model overall is displayed in bold. Subscripts denote standard deviation.

| Dataset | Labels/ Class | GCN | Co-train | Union | Intersection | Self-train | FASTRAIN -GCN | Self-train -CaGCN | FASTRAIN -CaGCN |
|---|---|---|---|---|---|---|---|---|---|
| Cora | 1 | $48.9_{2.4}$ | $53.3_{2.0}$ | $53.6_{2.2}$ | $51.8_{1.6}$ | $52.7_{2.2}$ | $\underline{56.9_{1.9}}$ | $57.2_{1.9}$ | $\mathbf{61.6_{2.1}}$ |
| | 2 | $64.3_{1.8}$ | $66.5_{1.9}$ | $66.3_{2.2}$ | $64.2_{1.7}$ | $66.1_{1.9}$ | $\underline{68.8_{1.9}}$ | $69.5_{1.4}$ | $\mathbf{73.0_{1.5}}$ |
| | 4 | $69.1_{2.1}$ | $71.1_{1.6}$ | $71.2_{1.5}$ | $69.7_{1.3}$ | $70.8_{1.7}$ | $\underline{72.7_{1.5}}$ | $71.9_{1.3}$ | $\mathbf{74.1_{1.2}}$ |
| | 8 | $74.8_{0.9}$ | $75.0_{0.8}$ | $75.7_{0.9}$ | $74.7_{0.9}$ | $75.5_{0.7}$ | $\underline{77.4_{0.8}}$ | $79.4_{0.6}$ | $\mathbf{81.6_{0.5}}$ |
| | 16 | $78.8_{1.1}$ | $78.9_{0.8}$ | $78.8_{0.7}$ | $78.2_{0.7}$ | $79.1_{0.8}$ | $\underline{80.6_{0.8}}$ | $82.2_{0.6}$ | $\mathbf{83.4_{0.6}}$ |
| Citeseer | 1 | $38.1_{3.6}$ | $39.8_{3.3}$ | $39.5_{3.1}$ | $38.2_{2.9}$ | $39.3_{3.4}$ | $\underline{43.1_{3.5}}$ | $46.2_{2.9}$ | $\mathbf{53.1_{3.1}}$ |
| | 2 | $49.9_{3.2}$ | $52.4_{2.5}$ | $52.6_{2.9}$ | $52.6_{2.6}$ | $52.9_{2.6}$ | $\underline{54.8_{2.8}}$ | $53.9_{2.4}$ | $\mathbf{57.1_{2.4}}$ |
| | 4 | $58.1_{2.3}$ | $60.4_{2.2}$ | $60.3_{2.5}$ | $60.7_{2.2}$ | $60.1_{2.5}$ | $\underline{63.8_{2.6}}$ | $64.7_{1.9}$ | $\mathbf{68.2_{1.7}}$ |
| | 8 | $64.4_{1.4}$ | $66.2_{1.6}$ | $65.9_{1.7}$ | $66.0_{1.6}$ | $66.8_{1.5}$ | $\underline{68.9_{1.3}}$ | $70.6_{1.1}$ | $\mathbf{72.8_{1.1}}$ |
| | 16 | $69.1_{1.5}$ | $70.2_{1.1}$ | $70.1_{0.9}$ | $70.3_{1.1}$ | $70.0_{1.1}$ | $\underline{71.6_{1.1}}$ | $72.1_{0.8}$ | $\mathbf{73.1_{0.6}}$ |
| Pubmed | 1 | $42.8_{2.5}$ | $58.2_{2.7}$ | $57.9_{2.5}$ | $56.8_{2.8}$ | $57.7_{2.7}$ | $\underline{60.4_{2.9}}$ | $66.5_{2.3}$ | $\mathbf{71.4_{2.1}}$ |
| | 2 | $56.1_{2.2}$ | $68.2_{2.1}$ | $68.6_{2.8}$ | $67.9_{2.2}$ | $68.4_{2.7}$ | $\underline{70.6_{2.6}}$ | $69.6_{2.5}$ | $\mathbf{73.9_{2.1}}$ |
| | 4 | $63.3_{1.7}$ | $69.4_{2.1}$ | $69.6_{2.4}$ | $69.6_{2.1}$ | $69.7_{1.9}$ | $\underline{71.1_{2.1}}$ | $70.9_{1.6}$ | $\mathbf{74.3_{1.9}}$ |
| | 8 | $69.9_{1.6}$ | $71.4_{1.1}$ | $71.3_{0.8}$ | $71.0_{1.1}$ | $71.7_{1.4}$ | $\underline{72.9_{1.3}}$ | $74.3_{0.7}$ | $\mathbf{76.2_{1.0}}$ |
| | 16 | $76.7_{0.6}$ | $77.2_{1.1}$ | $77.4_{0.9}$ | $77.5_{0.4}$ | $77.4_{0.4}$ | $\underline{78.6_{0.6}}$ | $78.1_{0.7}$ | $\mathbf{79.4_{0.5}}$ |
| CoraFull | 1 | $26.4_{3.8}$ | $30.4_{2.9}$ | $30.1_{4.1}$ | $27.3_{3.3}$ | $29.1_{3.5}$ | $\underline{31.1_{3.8}}$ | $30.6_{3.7}$ | $\mathbf{31.8_{3.9}}$ |
| | 2 | $29.6_{4.1}$ | $32.8_{3.8}$ | $33.1_{3.9}$ | $29.7_{3.2}$ | $33.0_{4.3}$ | $\underline{35.1_{4.0}}$ | $33.8_{2.9}$ | $\mathbf{35.6_{3.3}}$ |
| | 4 | $43.2_{2.4}$ | $44.8_{2.0}$ | $44.9_{3.1}$ | $43.9_{2.6}$ | $44.7_{2.1}$ | $\underline{46.6_{2.8}}$ | $45.9_{3.0}$ | $\mathbf{47.2_{2.8}}$ |
| | 8 | $53.2_{2.6}$ | $55.0_{1.9}$ | $55.2_{2.2}$ | $54.6_{2.0}$ | $55.6_{2.2}$ | $\underline{56.4_{1.9}}$ | $56.3_{1.6}$ | $\mathbf{57.2_{1.8}}$ |

### 4.1 Primary Results

We present results on the Cora, Citeseer, Pubmed and CoraFull datasets with different label rates using different GNN architectures – GCN in Table 2 and GAT in Table 3. To provide a fair comparison of FASTRAIN-GNN with other methods (except CaGCN), we ensure the following – (1) the same set of nodes are labeled for all training methods. (2) The confidence threshold $T_c$ is fixed at 0.8. (3) 4 stages of self-training are performed, with 500 epochs of training in each stage in all methods. For all experiments on CaGCN, we follow the best hyperparameter settings described by the authors (Wang et al., 2021). We note that our implementation of the baseline self-training methods (Co-training, Union, Intersection, Self-training) achieves significantly higher accuracy (by up to 10%) than previously reported (Li et al., 2018; Zhou et al., 2019). This is because we optimize the baselines (see Appendix B) by choosing the optimal

number of GNN layers for different label rates, and by randomly initializing the GNN model at the start of each stage of self-training (rather than initializing with the trained weights from the previous stage). We find that FASTRAIN-GNN consistently outperforms conventional self-training methods under different label rates (Tables 2, 3, 7). **In addition to producing models that are up to** $4.2\%$ **more accurate than these optimized baselines, FASTRAIN-GNN also accelerates the self-training process, reducing the wall-clock self-training time by** $\{1.7\times, 1.9\times, 2.1\times$ **and** $1.7\times\}$ **on** $\{$**Cora, Citeseer, Pubmed and CoraFull**$\}$ **respectively compared to conventional self-training**. In addition, we demonstrate that FASTRAIN-GNN can be used in conjunction with the current state-of-the-art method (to the best of our knowledge) for self-training – CaGCN (Wang et al., 2021). This combination, which we call FASTRAIN-CaGCN and FASTRAIN-CaGAT, uses confidence calibration to improve the quality of pseudolabels generated in each stage to achieve further accuracy gains of up to $4.4\%$ using the same hyperparameters and initial set of labeled nodes.

Table 3: **Results of training GAT with different label rates.** The most accurate model without confidence calibration is underlined, while the most accurate model overall is displayed in bold. Subscripts denote standard deviation.

| Dataset | Labels/ Class | GAT | Co-train | Union | Intersection | Self-train | FASTRAIN -GAT | Self-train -CaGAT | FASTRAIN -CaGAT |
|---|---|---|---|---|---|---|---|---|---|
| Cora | 1 | $49.7_{2.7}$ | $54.8_{2.9}$ | $54.5_{2.3}$ | $53.6_{2.5}$ | $53.9_{2.6}$ | $\underline{57.1}_{2.6}$ | $59.3_{2.5}$ | $\mathbf{63.3}_{2.4}$ |
|  | 2 | $65.1_{2.9}$ | $67.6_{2.3}$ | $67.7_{1.9}$ | $66.4_{1.7}$ | $66.3_{1.9}$ | $\underline{69.6}_{2.0}$ | $70.4_{2.2}$ | $\mathbf{74.1}_{1.8}$ |
|  | 4 | $70.2_{1.7}$ | $71.8_{1.6}$ | $71.6_{2.3}$ | $70.6_{2.1}$ | $71.9_{1.5}$ | $\underline{73.2}_{1.9}$ | $72.9_{1.4}$ | $\mathbf{75.8}_{1.4}$ |
|  | 8 | $75.4_{1.8}$ | $76.2_{1.4}$ | $76.7_{1.4}$ | $75.0_{1.0}$ | $76.3_{1.3}$ | $\underline{78.4}_{1.6}$ | $79.8_{1.6}$ | $\mathbf{81.7}_{1.0}$ |
|  | 16 | $79.1_{0.9}$ | $79.4_{1.5}$ | $79.8_{1.3}$ | $78.8_{0.7}$ | $80.0_{0.8}$ | $\underline{82.1}_{0.6}$ | $83.0_{1.0}$ | $\mathbf{84.2}_{0.7}$ |
| Citeseer | 1 | $36.9_{4.1}$ | $38.7_{4.3}$ | $38.8_{4.7}$ | $37.7_{3.9}$ | $38.3_{4.4}$ | $\underline{40.2}_{4.6}$ | $44.8_{4.2}$ | $\mathbf{48.9}_{3.9}$ |
|  | 2 | $49.0_{3.8}$ | $51.6_{3.7}$ | $51.7_{3.7}$ | $51.9_{4.3}$ | $51.7_{3.5}$ | $\underline{53.6}_{3.7}$ | $52.5_{3.9}$ | $\mathbf{55.8}_{3.7}$ |
|  | 4 | $56.4_{2.9}$ | $58.8_{3.1}$ | $58.2_{3.3}$ | $59.1_{2.7}$ | $59.2_{3.0}$ | $\underline{61.9}_{3.4}$ | $62.3_{3.4}$ | $\mathbf{65.7}_{2.9}$ |
|  | 8 | $62.8_{1.8}$ | $64.9_{2.6}$ | $64.3_{2.7}$ | $65.2_{2.2}$ | $65.2_{2.2}$ | $\underline{66.6}_{1.9}$ | $67.4_{2.1}$ | $\mathbf{69.5}_{2.1}$ |
|  | 16 | $66.8_{2.1}$ | $68.7_{1.7}$ | $68.8_{2.0}$ | $69.3_{1.5}$ | $69.0_{1.8}$ | $\underline{70.2}_{1.5}$ | $70.8_{1.5}$ | $\mathbf{72.0}_{1.7}$ |
| Pubmed | 1 | $40.7_{2.8}$ | $55.4_{2.6}$ | $55.9_{2.7}$ | $54.8_{2.7}$ | $54.5_{2.4}$ | $\underline{57.2}_{2.8}$ | $59.9_{2.4}$ | $\mathbf{63.7}_{2.6}$ |
|  | 2 | $52.6_{2.1}$ | $64.7_{2.9}$ | $64.6_{2.3}$ | $63.8_{2.6}$ | $65.1_{2.7}$ | $\underline{68.4}_{2.2}$ | $68.6_{2.1}$ | $\mathbf{72.4}_{2.7}$ |
|  | 4 | $62.1_{2.3}$ | $68.4_{1.9}$ | $68.1_{2.1}$ | $68.6_{1.8}$ | $68.5_{1.8}$ | $\underline{70.3}_{2.1}$ | $70.1_{2.3}$ | $\mathbf{73.6}_{1.8}$ |
|  | 8 | $69.0_{1.7}$ | $70.9_{1.7}$ | $71.2_{1.5}$ | $71.1_{1.6}$ | $71.4_{1.8}$ | $\underline{72.5}_{1.6}$ | $73.7_{1.4}$ | $\mathbf{75.6}_{1.7}$ |
|  | 16 | $76.3_{0.7}$ | $76.9_{1.1}$ | $77.0_{1.0}$ | $77.1_{0.7}$ | $77.2_{0.7}$ | $\underline{78.4}_{0.7}$ | $77.9_{0.4}$ | $\mathbf{79.0}_{0.7}$ |
| CoraFull | 1 | $28.6_{4.4}$ | $31.8_{4.7}$ | $32.1_{3.9}$ | $29.7_{4.3}$ | $31.6_{3.9}$ | $\underline{32.7}_{4.2}$ | $33.1_{3.7}$ | $\mathbf{34.6}_{4.2}$ |
|  | 2 | $31.2_{3.7}$ | $33.7_{4.5}$ | $34.2_{4.7}$ | $32.9_{3.7}$ | $33.8_{4.6}$ | $\underline{35.5}_{4.7}$ | $34.2_{3.9}$ | $\mathbf{36.7}_{3.6}$ |
|  | 4 | $45.4_{3.2}$ | $47.2_{3.3}$ | $47.4_{4.2}$ | $46.6_{2.9}$ | $46.9_{2.9}$ | $\underline{48.3}_{3.3}$ | $47.7_{3.5}$ | $\mathbf{49.0}_{3.7}$ |
|  | 8 | $53.4_{2.7}$ | $55.6_{2.1}$ | $55.7_{2.9}$ | $55.0_{2.2}$ | $55.7_{2.5}$ | $\underline{56.5}_{2.2}$ | $56.2_{2.1}$ | $\mathbf{57.4}_{2.3}$ |

## 4.2 Ablation Study: Breakdown of benefits from different FASTRAIN-GNN optimizations

We perform an ablation study to evaluate the impact of each optimization performed by the FASTRAIN-GNN framework. The results are presented in Table 4. SPF improves accuracy by removing nodes whose pseudolabels are likely to be incorrect from the training set in each stage, and it improves efficiency by reducing the number of training samples in each stage. DS and DR improve accuracy by taking advantage of the layer effect in GNNs (Sun et al.), and by reducing the impact of incorrectly labeled nodes in the training set, respectively. In addition, they improve efficiency by training smaller GNNs on sparser graphs in later stages. Finally, PGP improves accuracy by reducing noise from inter-class edges (Chen et al., 2020a), and it improves efficiency by making the graphs more sparse. We also note that the different optimizations add minimal overheads (Appendix E). In the following subsections, we discuss and analyze the contribution of each FASTRAIN-GNN optimization in improving the accuracy and efficiency of the previous state-of-the-art self-training method (CaGCN).

### 4.2.1 SPF improves the quality of generated pseudolabels.

Confidence-based pseudolabeling can lead to noisy labels, even when the models are confidence-calibrated, as seen in Fig. 7. By adding a second layer of filtering based on robustness to small perturbations in the graph structure, SPF consistently generates more accurate pseudolabels than confidence-based pseudolabeling.

Table 4: **Accuracy and efficiency gains from the different FASTRAIN-GNN optimizations on Cora.** Speedup is computed over self-training of the more accurate among 2/3-layer models, and averaged across different train/test splits and label rates.

| Labels/Class | Self-Training-CaGCN (2-layer/ 3-layer CaGCN) | with SPF and DR (2-layer/ 3-layer CaGCN) | with SPF, DR and DS | with SPF, DR, DS and PGP |
|---|---|---|---|---|
| 1 | $55.8_{2.1}$ / $57.2_{1.9}$ | $57.1_{2.3}$ / $58.3_{1.8}$ | $60.4_{1.9}$ | $61.6_{2.1}$ |
| 2 | $64.4_{1.1}$ / $69.5_{1.4}$ | $66.1_{1.7}$ / $70.9_{1.6}$ | $72.2_{1.7}$ | $73.0_{1.5}$ |
| 4 | $68.2_{1.7}$ / $71.9_{1.3}$ | $69.1_{1.8}$ / $72.6_{1.6}$ | $73.4_{1.1}$ | $74.1_{1.2}$ |
| 8 | $79.4_{0.6}$ / $77.6_{1.1}$ | $80.3_{1.1}$ / $78.9_{0.9}$ | $80.9_{0.8}$ | $81.6_{0.5}$ |
| 16 | $82.2_{0.6}$ / $80.3_{0.6}$ | $82.7_{0.4}$ / $81.5_{0.5}$ | $82.7_{0.5}$ | $83.4_{0.6}$ |
| **Average Speedup** | 1X | 1.3X | 1.6X | 1.7X |

While the quality of confidence-based pseudolabels can be improved by increasing the confidence threshold, we demonstrate that SPF produces (1) more accurate pseudolabels when both methods add the same number of nodes to the training set, and (2) a smaller set of training nodes with more accurate pseudolabels for different confidence thresholds in Fig. 7. In fact, with a confidence threshold of 0.99, SPF generates training sets that are 1.2× smaller with 5.3% more accurate labels on Cora (4 labeled nodes per class) during self-training. Similarly, with 8 labeled nodes per class, SPF generates training sets that are 1.2× smaller with 4.8% more accurate labels compared to training sets generated solely based on confidence. We find that using 10 $spf\_iterations$ with a $spf\_sampling\_rate$ of 0.9 works well in all our experiments. While the accuracy of training labels can be marginally improved by using smaller values of $spf\_sampling\_rate$ (Fig 4), the number of nodes added to the training set in each stage decreases greatly. For instance, on Cora with 4 labels per class, 1801 nodes are added to the training set (label accuracy = 85.1%) over the course of self-training with a $spf\_sampling\_rate$ of 0.9. When a $spf\_sampling\_rate$ of 0.8 is used, only 1206 nodes are added to the training set (label accuracy = 85.4%). As a result, the accuracy gains from multi-stage self-training are diminished when $spf\_sampling\_rate$ is too small.

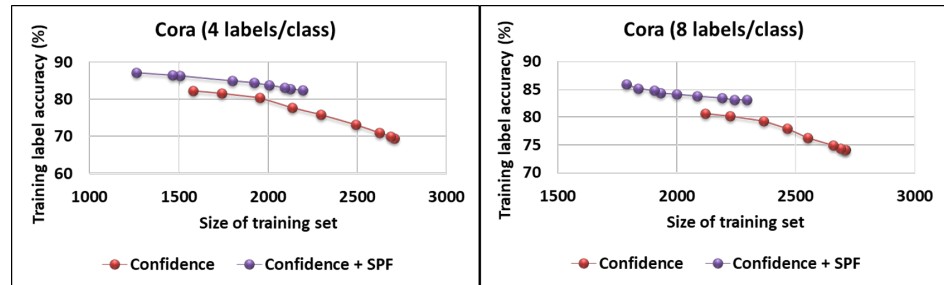

Figure 7: **Accuracy of pseudolabels generated for different confidence thresholds with and without SPF.** Results shown are based on all the nodes added to the training set during self-training of a 3-layer CaGCN model. The different points on the curves are obtained by varying the confidence threshold from 0.2 (far-right) to 0.99 (far-left). For SPF, 10 $spf\_iterations$ are used with a $spf\_sampling\_rate$ of 0.9 to test consistency of predictions.

### 4.2.2 DR reduces the impact of noisy training labels.

While SPF helps reduce noise in training labels by filtering out nodes that are likely to be incorrectly classified, DR minimizes the impact of residual errors in training labels after SPF. We demonstrate that dynamically altering the sampling rate is effective at reducing the impact of label noise in Fig. 8. During the first stage of self-training (when golden labels are used), higher sampling rates provide better accuracy. As label accuracy decreases, lower sampling rates prevent overfitting to noisy labels. Therefore, we find that reducing the sampling rate after the first stage of self-training leads to significant improvement in accuracy of the final self-trained model (Fig. 8), and we empirically find that reducing the sampling rate from 0.9 (10% of incident edges of node are randomly pruned in each training epoch) in the first stage to 0.8 after the first stage provides highest accuracy across datasets and GNN architectures.

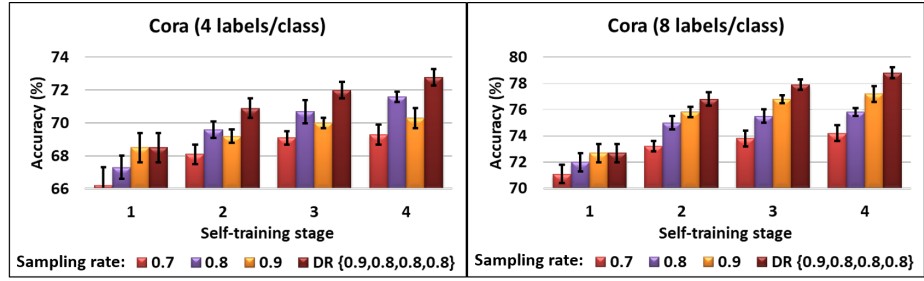

Figure 8: **CaGCN accuracy across self-training stages with different sampling rates.** Results are obtained from self-training a 3-layer CaGCN model with confidence threshold 0.8.

### 4.2.3 DS overcomes the layer effect across different self-training stages to improve final accuracy.

Prior research has demonstrated the "layer effect" in GNNs, where deeper GNNs are required for effective information propagation in the presence of very few training nodes (Sun et al.). As the number of training nodes increases, GNNs with less layers tend to achieve higher accuracy due to the over-smoothing problem (Chen et al., 2020a) in deeper GNNs. Here, we demonstrate that the "layer effect" plays a significant role during the different stages of self-training (Fig. 9). With 4 labeled nodes per class on Cora, the 3-layer CaGCN outperforms the 2-layer CaGCN due to better propagation of label information. However, we demonstrate that substantial accuracy gains can be achieved by switching to a 2-layer CaGCN model after the first stage of self-training when more nodes are added to the training set (approximately 300 nodes per class in this case). When starting with 8 labeled nodes per class, the 2-layer CaGCN model achieves higher accuracy at the end of self-training, even though a 3-layer CaGCN model is more accurate (and hence, generates more accurate pseudolabels) after the first stage. Since the accuracy gap between the models at the end of the first stage is small, we find that the over-smoothing issue outweighs the advantages of a more accurate training set in the 3-layer CaGCN model. Therefore, we find that using a 3-layer CaGCN model in the first stage of self-training and a 2-layer CaGCN model in all subsequent stages enables reaping the benefits of a more accurate training set without encountering the over-smoothing problem, leading to the most accurate predictions on unlabeled nodes. Across different datasets and GNN architectures, we empirically observe that when approximately 2.5% of all nodes in the graph are labeled, there is sufficient supervision for 2-layer models to outperform 3-layer (and deeper) models in all stages of self-training. As a result, when number of labeled nodes per class increases, the accuracy gains from DS diminish, since there is sufficient supervision to cause over-smoothing in deep GNNs even in the first self-training stage (Table 4).

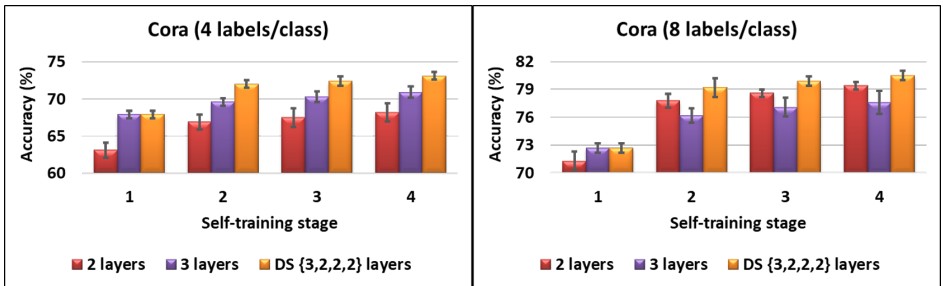

Figure 9: **Stage-wise accuracy during self-training of CaGCN with confidence threshold of 0.8.**

### 4.2.4 PGP significantly outperforms single-shot pruning of inter-class edges.

PGP only prunes inter-class edges between nodes in the training set in each stage. PGP improves accuracy over methods that do not prune inter-class edges, since inter-class edges add noise and exacerbate the over-smoothing problem (Fig. 10). When only few training nodes are available, PGP also significantly outperforms one-shot pruning (edges are pruned based on predictions of all nodes after the first stage of self-training; if the prediction of a node does not match the prediction of its neighbor, then the edge between the node and its neighbor is pruned). In fact, the difference in accuracy between PGP and one-shot pruning can be as high as 15%, and this difference is highest when the number of labeled nodes/class is minimum (Fig. 10). In fact, we find that with 4 labels/class, one-shot pruning prunes 54% of all inter-class edges in the graph, while also incorrectly pruning 21% of all intra-class edges, thereby leading to a large drop in accuracy. In contrast, PGP prunes 62% of all inter-class edges, while pruning only 7% of all intra-class edges at the end of self-training.

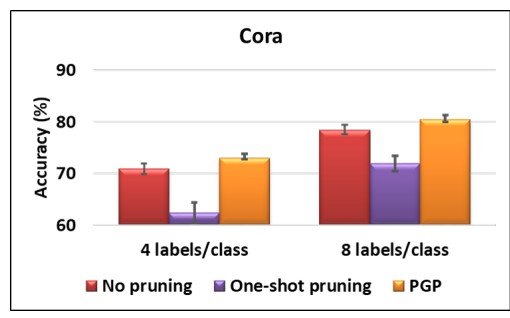

Figure 10: **CaGCN accuracy with different pruning methods.** Results are obtained from self-training a 3-layer CaGCN model with confidence threshold of 0.8.

## 5 Discussion

In this section, we describe some limitations of the proposed framework and identify directions for future research.

While SPF, DR and DS work well on homophilous and heterophilous graphs, pruning inter-class edges through PGP is applicable only to homophilous graphs, where connected nodes are expected to have similar labels (for example, citation networks, where authors are likely to cite authors working in similar fields). In the case of heterophilous graphs, connected nodes have dissimilar labels, and hence, pruning inter-class edges is highly undesirable. In graphs that rely on the heterophily assumption, a slightly modified version of PGP – pruning intra-class edges instead of inter-class edges – leads to accuracy gains by reducing the effect of over-smoothing (see Appendix F). However, PGP cannot be trivially modified to work with graphs where both inter-class and intra-class edges are prominent (for example, molecular graphs, where atoms are connected to both other atoms of the same kind, and to atoms of different kinds).

In FASTRAIN-GNN, we only consider optimizations that simultaneously improve both accuracy and computational efficiency. Future work can use insights from FASTRAIN-GNN to maximize either the accuracy or training efficiency of self-training but not both. For instance, adding virtual intra-class edges (in addition to pruning inter-class edges with PGP) can further improve the information-to-noise during message passing, and hence, further increase accuracy at the cost of training efficiency. Similarly, pruning less-important edges also during PGP (using methods such as Lottery Ticket Hypothesis (Chen et al., 2021)) can lead to further efficiency gains at a small accuracy cost.

## 6 Conclusion

In this work, we presented FASTRAIN-GNN, a framework for fast and accurate self-supervised training of GNNs. The framework introduced four major optimizations to conventional multi-stage self-training. Sampling-based Pseudolabel Filtering was proposed to obtain smaller training sets with more accurate training labels. Dynamic Regularization was proposed to reduce the impact of residual errors in training labels after SPF. Dynamic Sizing was introduced to overcome the layer effect during self-training. Finally, Progressive Graph Pruning incrementally removed inter-class edges from the graph to improve the information-to-noise ratio during message passing. We demonstrated that FASTRAIN-GNN simultaneously accelerates the self-training process and improves the accuracy of the self-trained models over the current state-of-the-art methods.

## 7 Acknowledgement

This work was supported in part by the Center for the Co-Design of Cognitive Systems (CoCoSys), a JUMP2.0 center sponsored by the Semiconductor Research Corporation (SRC) and DARPA, and in part by SRC under the AIHW program.

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

## A    Computational demands of conventional self-training

Conventional self-training can lead to substantial accuracy gains (up to 14.9% absolute gain) over supervised training, but it comes at a large computational cost (Table 5). We find that 4-stage self-training always takes $> 4\times$ longer than single-stage supervised training on only the labeled nodes for two reasons: (1) each stage of self-training is performed on a progressively larger training set, and (2) training set augmentation also adds some overheads, since all unlabeled nodes need to be predicted (see Appendix E). As the number of labeled nodes increases, the accuracy gain from self-training decreases, since there is sufficient supervision to learn effectively from only the labeled nodes. We also analyze the impact of the number of stages of self-training on both accuracy and wall-clock training time (Fig. 11). Initially, as the number of self-training stages, the accuracy of the self-trained model also increases. However, when the number of stages is increased beyond a certain number, the accuracy gain saturates, while the wall-clock training time continues to increase. We observe that this accuracy saturation happens because very few nodes are added to the training set in the later stages of self-training.

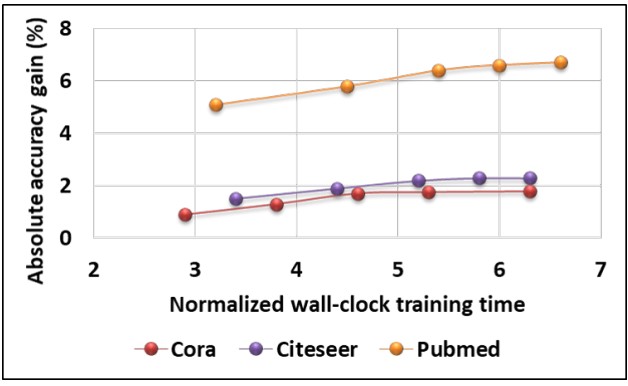

Figure 11: **Impact of number of stages of self-training.** Gains in accuracy (wall-clock training time) are reported over (normalized to) training using only the labeled nodes (4 labeled nodes/class).

## B    Optimized Baselines

We present two simple optimizations to the conventional self-training loop that improves the accuracy of previously reported baselines by up to 10%.

Table 5: **Benefits and overheads of conventional self-training.** Results are obtained from 4-stage self-training of a GCN model with all hyperparameters set to their optimal value. Gains in accuracy (absolute points)/ wall-clock training time are reported over/ normalized to supervised training using only the labeled nodes, and averaged across 100 random train-test splits.

| Dataset | Labeled nodes/class | Accuracy gain | Normalized training time |
|---------|---------------------|---------------|--------------------------|
|         | 1                   | 3.8           | 4.9                      |
|         | 2                   | 1.8           | 4.7                      |
| Cora    | 4                   | 1.7           | 4.6                      |
|         | 8                   | 0.7           | 4.4                      |
|         | 16                  | 0.3           | 4.1                      |
|         | 1                   | 1.2           | 5.7                      |
|         | 2                   | 3.0           | 5.4                      |
| Citeseer| 4                   | 2.0           | 5.2                      |
|         | 8                   | 2.4           | 4.9                      |
|         | 16                  | 0.9           | 4.8                      |
|         | 1                   | 14.9          | 5.5                      |
|         | 2                   | 12.4          | 5.5                      |
| Pubmed  | 4                   | 6.4           | 5.4                      |
|         | 8                   | 1.8           | 5.2                      |
|         | 16                  | 0.8           | 5.0                      |

### B.1 GNN model architecture selection for different label rates

At low label rates, deeper GNNs are more accurate due to better propagation of label information. At higher label rates, deeper GNNs suffer from the over-smoothing problem, where nodes belonging to different classes acquire very similar representations. As a result, shallow GNNs outperform deeper GNNs at higher label rates. Therefore, choosing the appropriate GNN architecture for the given label rate is vital, and leads to substantial accuracy improvements over previously reported baselines that always use 2-layer GNNs (Table 6).

Table 6: **Results of training GCN with different label rates.** GCN-opt chooses the appropriate GCN architecture for the given label rate. Self-training initializes the GCN model in each stage using the trained weights from the previous stage (random in stage 1), while Self-training-opt randomly initializes the GCN model in all stages. Both versions of self-training train the GCN-opt model.

| Dataset | Labels/Class | 2-layer GCN | GCN-opt | Self-training | Self-training-opt |
|---------|--------------|-------------|---------|---------------|-------------------|
|         | 1            | 41.1        | 48.9    | 51.3          | 52.7              |
|         | 2            | 60.7        | 64.3    | 65.0          | 66.1              |
| Cora    | 4            | 67.2        | 69.1    | 69.6          | 70.8              |
|         | 8            | 74.6        | 74.8    | 75.1          | 75.5              |
|         | 16           | 78.8        | 78.8    | 78.9          | 79.1              |
|         | 1            | 34.8        | 38.1    | 38.6          | 39.3              |
|         | 2            | 40.7        | 49.9    | 50.7          | 52.9              |
| Citeseer| 4            | 51.1        | 58.1    | 58.8          | 60.1              |
|         | 8            | 64.2        | 64.4    | 65.6          | 66.8              |
|         | 16           | 69.1        | 69.1    | 69.4          | 70.0              |
|         | 1            | 40.7        | 42.8    | 49.8          | 57.7              |
|         | 2            | 54.5        | 56.1    | 60.3          | 68.4              |
| Pubmed  | 4            | 62.1        | 63.3    | 66.2          | 69.7              |
|         | 8            | 69.4        | 69.9    | 70.5          | 71.7              |
|         | 16           | 76.7        | 76.7    | 77.1          | 77.4              |
|         | 1            | 25.8        | 26.4    | 28.2          | 29.1              |
|         | 2            | 27.4        | 29.6    | 31.8          | 33.0              |
| CoraFull| 4            | 41.1        | 43.2    | 43.7          | 44.7              |
|         | 8            | 53.0        | 53.2    | 54.8          | 55.6              |

### B.2 Random initialization of the GNN model in each stage

The majority of previous works (such as Zhou et al. (2019); Li et al. (2018) etc.) initialize the model using the trained weights from the previous stage (when stage > 1), and randomly initialize the weights only in stage 1. However, we find that randomly initializing the weights in all stages leads to more accurate models at the end of self-training (Fig. 12). In particular, we find that the total training loss is very small when weights are initialized from the trained model in the previous stage (even after the training set is enlarged), since only nodes with highly confident predictions are added to the training set. As a result, multi-stage self-training only leads to a small improvement in accuracy over single-stage training. In contrast, randomly initializing the model in each stage enables the model to make the best use of the enlarged training set (by effectively forgetting everything learnt in the previous stage, and learning on the enlarged training set from scratch, leading to better node representations). However, we also note that random initialization in all stages requires a larger number of training epochs in each stage for convergence, and hence, incurs larger training overheads. In our experiments, we find that the number of training epochs needed for convergence in each stage is approximately 50-100 with random initialization, and 25-50 when initializing from the previous stage.

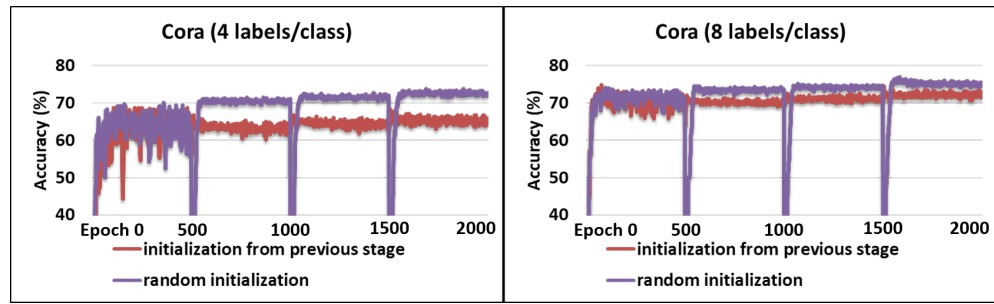

Figure 12: **Accuracy with and without random initialization in all stages.** Accuracy is measured on all unlabeled nodes in the graph (3-layer GCN model, 4-stage self-training, 500 training epochs per stage, confidence threshold = 0.8).

## C Comparison of previously proposed self-training methods

Table 7: **Results of training GCN with different self-training methods on Cora.** The most accurate model without using our FASTRAIN-GNN framework is underlined, while the most accurate model overall is displayed in bold.

| Self-training method | Accuracy of self-trained model | | |
|---|---|---|---|
| | 2 Labels/Class | 4 Labels/Class | 8 Labels/Class |
| Co-train (Li et al., 2018) | 66.5 | 71.1 | 75.0 |
| Union (Li et al., 2018) | 66.3 | 71.2 | 75.7 |
| Intersection (Li et al., 2018) | 64.2 | 69.7 | 74.7 |
| Self-train (Li et al., 2018) | 66.1 | 70.8 | 75.5 |
| M3S (Sun et al.) | 61.5 | 67.2 | 75.6 |
| DSGCN (Zhou et al., 2019) | 68.9 | 72.6 | 77.6 |
| CGCN (Hui et al., 2020) | 64.3 | 72.4 | 76.8 |
| Self-train-CaGCN (Wang et al., 2021) | 69.5 | 71.9 | 79.4 |
| FASTRAIN-CaGCN (Ours) | **73.0** | **74.1** | **81.6** |

While the the optimizations proposed in FASTRAIN-GNN can be used in conjunction with different pseudolabeling methods, we demonstrate that FASTRAIN-CaGCN produces more accurate models compared to previously proposed self-training methods (Table 7). For M3S and CGCN, we report results directly from

the respective papers. For all other methods, we report the average accuracy across 100 different random train-test splits.

## D    Hyperparameter Tuning

We describe how the different hyperparameters introduced in the FASTRAIN-GNN framework can be tuned to maximize the accuracy of the self-trained models. We note that PGP does not involve any hyperparameters. While the exact hyperparameters used in our experiments are listed in Table 8, we describe insights from our hyperparameter-tuning experiments in the following subsections to enable faster hyperparameter optimization in future works.

Table 8: **Hyperparameters used in our experiments.** The exact same hyperparameters are used in all our experiments spanning different datasets, GNN architectures and label rates.

| Hyperparameter | Description | Value |
|---|---|---|
| $spf\_sampling\_rate$ (SPF) | Fraction of edges retained in the graph in each sampled configuration | 0.9 |
| $spf\_iterations$ (SPF) | Number of sampled configurations generated for testing robustness of predictions | 10 |
| $sampling\_rate\_init$ (DR) | Sampling rate used in the first stage of self-training | 0.9 |
| $sampling\_rate\_final$ (DR) | Sampling rate used from the second stage of self-training onward | 0.8 |
| $deep\_gnn$ (DS) | Architecture of the deep GNN used in the initial stages of self-training | 3-layer GCN/ GAT |
| $shallow\_gnn$ (DS) | Architecture of the shallow GNN used in the later stages of self-training | 2-layer GCN/ GAT |
| $DS\_Threshold$ (DS) | The size of the training set at which the $deep\_gnn$ is to be replaced by the $shallow\_gnn$ | >=2.5% of nodes in the graph are labeled or pseudolabeled |

### D.1    SPF hyperparameters

SPF uses two hyperparameters. (1) $spf\_sampling\_rate$ denotes the fraction of edges that are retained in the graph in each sampled configuration used to test the robustness of predictions, and (2) $spf\_iterations$ denotes the number of sampled configurations that are used to test the robustness of predictions (only those nodes whose predictions remain constant across all sampled configurations are added to the training set).

Table 9: **Impact of $spf\_sampling\_rate$ on accuracy of the self-trained model.** Results are obtained from self-training a CaGCN model using different $spf\_sampling\_rates$ (all other hyperparameters are set to their optimal value).

| Dataset | $spf\_sampling\_rate$ | Number of nodes added to training set | Accuracy of nodes added to training set | Accuracy of self-trained model |
|---|---|---|---|---|
| Cora (4 labels/class) | 0.95 | 2194 | 83.9 | 73.2 |
| | 0.9 | 1801 | 85.1 | 74.1 |
| | 0.85 | 1745 | 85.3 | 73.9 |
| | 0.8 | 1206 | 85.4 | 73.5 |
| | 0.7 | 917 | 85.6 | 73.3 |
| Cora (16 labels/class) | 0.95 | 2421 | 83.1 | 82.7 |
| | 0.9 | 2168 | 87.3 | 83.4 |
| | 0.85 | 1947 | 87.8 | 83.4 |
| | 0.8 | 1601 | 88.1 | 82.9 |
| | 0.7 | 1105 | 88.4 | 82.7 |

When $spf\_sampling\_rate$ is too large (>=0.95), we find that there is insufficient perturbation to the graph structure to distinguish between robust and non-robust predictions. As a result, the predictions of the vast

majority of candidate nodes remain constant across sampled configurations, and SPF becomes ineffective (Table 9). On the other hand, when $spf\_sampling\_rate$ is too small ($<=0.8$), the number of nodes with consistent predictions decreases drastically. The primary reason for the high effectiveness of self-training for few-shot transductive node classification is the increased number of nodes taking part in the training process through training set augmentation (either because they are added to the training set, or because they are neighbors of nodes added to the training set). As a result, the decreased coverage of nodes taking part in training with a small $spf\_sampling\_rate$ outweighs the benefits of more accurate training labels (Table 9). In our experiments, we find that $spf\_sampling\_rate = 0.9$ provides the best trade-off between coverage of nodes seen during training and the accuracy of training labels (across all datasets and label rates).

Since sampling randomly prunes edges from the graph, some sampled configurations will not have any impact on a given node. As a result, $spf\_iterations$ should be sufficiently high to ensure that perturbations affect the computation graphs of all candidate nodes (Table 10). In our experiments, we find that $spf\_iterations = 10$ is sufficient to distinguish between robust and non-robust predictions (across all datasets and label rates). We note that the computational complexity of SPF is directly proportional to $spf\_iterations$, and statistically significant accuracy gains are not obtained with increasing $spf\_iterations$ beyond 10.

Table 10: **Impact of** $spf\_iterations$ **on accuracy of the self-trained model.** Results are obtained from self-training a CaGCN model using different $spf\_iterations$ (all other hyperparameters are set to their optimal value).

| Dataset | $spf\_iterations$ | Number of nodes added to training set | Accuracy of nodes added to training set | Accuracy of self-trained model |
|---|---|---|---|---|
| | 2 | 2068 | 84.2 | 73.4 |
| | 5 | 2008 | 84.6 | 73.8 |
| Cora (4 labels/class) | 10 | 1801 | 85.1 | 74.1 |
| | 20 | 1797 | 85.3 | 74.2 |
| | 50 | 1796 | 85.3 | 74.2 |
| | 2 | 2378 | 85.7 | 82.9 |
| | 5 | 2182 | 86.9 | 83.1 |
| Cora (16 labels/class) | 10 | 2168 | 87.3 | 83.4 |
| | 20 | 2167 | 87.3 | 83.4 |
| | 50 | 2164 | 87.4 | 83.4 |

### D.2  DR hyperparameters

DR uses two hyperparameters. (1) $sampling\_rate\_init$ denotes the sampling rate in the first stage of self training (when only nodes with golden labels are used for training), and (2) $sampling\_rate\_final$ denotes the sampling rate used when pseudolabeled nodes are added to the training set. We use $sampling\_rate\_init$ in the first stage of self-training, and $sampling\_rate\_final$ in all subsequent stages. Sampling in GNNs is analogous to dropout (Srivastava et al., 2014) in other classes of neural networks. Dropout (a commonly used regularizer for training DNNs) prunes a random subset of neurons in the model in each training iteration. Similarly, sampling prunes a random subset of edges in the graph in each training iteration. As a result, sampling provides a regularization effect that prevents overfitting to training labels, thereby improving generalization performance (Li et al., 2022). The benefits of sampling can also be seen in Fig. 3. A dropout rate of p indicates that the probability of a neuron being pruned in a given training iteration is (1-p). Similarly, a sampling rate of s also indicates that the probability of an edge in the graph being pruned in a given training iteration is (1-s).

In the first stage of self-training, the labels are 100% accurate. In addition, only a small number of training nodes are available. Therefore, we use a high value for $sampling\_rate\_init$ to maximize the number of nodes taking part in the training process, thereby enabling better propagation of information from labeled nodes. We find that $sampling\_rate\_init = 0.9$ works well across datasets and label rates in our experiments (Fig. 3). In fact, we observe that even in the presence of only golden labels, the regularization effect from sampling is important (when $sampling\_rate\_init >= 0.95$, there is an accuracy drop due to overfitting to the limited training data).

While sampling is an effective regularizer for preventing overfitting to noisy training labels, we find that under-sampling the graph adversely affects convergence during training, thereby preventing learning from correct labels also. We empirically find that $sampling\_rate\_final = 0.8$ works well across datasets and label rates in our experiments, since it provides the best balance between preventing overfitting and convergence. This can be seen in Fig. 3, where a sampling rate of 0.8 typically leads to the most accurate trained models when the accuracy of training labels is between 80% and 100%.

### D.3 DS hyperparameters

DS uses three hyperparameters. (1) $deep\_gnn$ denotes the architecture of the deep GNN used in the initial stages of self training. (2) $shallow\_gnn$ denotes the architecture of the shallow GNN used in the later stages of self training. (3) $DS\_Threshold$ determines the stage of self-training at which the $deep\_gnn$ is replaced by the $shallow\_gnn$.

The choice of architecture for both $deep\_gnn$ and $shallow\_gnn$ depends on the GNN model that is used for self-training, since different GNN variants require different numbers of layers for best accuracy. On the widely studied GCN and GAT models used in our experiments, 2-layer models have been reported to be the most accurate for supervised training on a large number of labeled nodes. We observe that 3-layer models are most accurate for few-shot learning. Therefore, we choose 3-layer GCN/ GAT models as $deep\_gnn$, and 2-layer GCN/ GAT models as $shallow\_gnn$.

When training with few labeled nodes, deep GNNs outperform shallow GNNs due to larger message passing coverage (enabling more unlabeled nodes to be "seen" during training). On the other hand, the increased message passing coverage leads to oversmoothing in deep GNNs when training on a large number of labeled nodes, resulting in shallow GNNs outperforming deep GNNs. The $DS\_Threshold$ is used to indicate the point at which oversmoothing outweighs the benefits of increased message passing coverage as the training set is augmented during self-training. We empirically observe that when approximately 2.5% of all nodes in the graph are labeled, there is sufficient coverage for $shallow\_gnns$ to outperform $deep\_gnns$ (Fig. 5), since >75% of all nodes in the graph take part in training using shallow GNNs (either because they are added to the training set, or they are neighbors of nodes in the training set). This holds across the different datasets, label rates and GNN architectures used in our experiments. We also find that this holds irrespective of the choice of initial training nodes. For instance, on Cora with 4 labeled nodes/class, the 3-layer GCN model achieves higher accuracy than the 2-layer GCN model at the end of the first stage of self-training in all 100 out of 100 trials with random train-test splits. Similarly, on Cora with 16 labeled nodes/class, the 2-layer GCN model achieves higher accuracy than the 3-layer GCN model at the end of the first stage of self-training in 98 out of 100 trials.

## E Overheads of the FASTRAIN-GNN optimizations

We find that the different FASTRAIN-GNN optimizations add minimal overheads (Fig. 13). SPF involves generating $spf\_iterations$ different sampled configurations (which takes only a few milliseconds), and predicting the high-confidence candidate nodes with different sampled configurations. We find that the fraction of runtime spent in SPF (8-10% of the total runtime) is similar to the fraction of runtime spent in computing the prediction confidence of all unlabeled nodes. This is because the additional computational complexity of predicting candidate nodes multiple times is amortized by two factors: (1) the set of candidate nodes is typically a small subset of all unlabeled nodes in the graph, and (2) each prediction is faster, since edges in the graph are pruned through sampling. DS and DR together account for <0.01% of the total runtime, since they only involve modifying hyperparameters. PGP involves identifying and pruning inter-class edges between labeled nodes. At

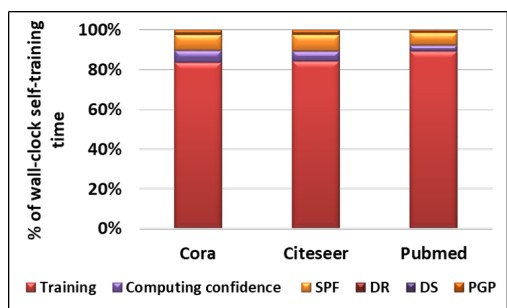

Figure 13: **Fraction of wall-clock self-training time spent in each step.** Results are obtained from self-training a 3-layer GCN model with confidence threshold of 0.8.

the end of each stage of self-training, PGP analyzes the labeled
neighbors of nodes that were added to the training set in that stage to identify inter-class edges. Since 1-hop neighbors of labeled nodes can be quickly identified from the adjacency matrix, PGP only adds small overheads, typically accounting for <2% of the total runtime.

## F    Results on heterophilous graphs

In this section, we present results on heterophilous graphs, where connected nodes are likely to have dissimilar labels. We follow the procedure described in GC-NII (Chen et al., 2020b) for processing the datasets. Similar to the experimental setup for homophilous graphs, we randomly select (labels/class) nodes of each class as training nodes, and report results

Table 11: **Dataset characteristics.** Homophily ratio is the ratio of edges connecting nodes of the same class to the total number of edges.

| Dataset | Classes | Nodes | Edges | Feature size | Homophily Ratio |
|---|---|---|---|---|---|
| Chameleon | 4 | 2277 | 36101 | 2325 | 0.23 |
| Texas | 5 | 183 | 309 | 1703 | 0.11 |
| Wisconsin | 5 | 251 | 499 | 1703 | 0.21 |
| Cornell | 5 | 183 | 295 | 1703 | 0.3 |

on the rest of the nodes in the graph. We repeat this process 100 times for each value of (labels/class), and all results reported in this section are averaged across 100 different training splits. The datasets used for testing are summarized in Table 11. Since some classes contain too few nodes to get a meaningful train-test split in the Texas, Wisconsin and Cornell datasets, we do not present results with 16 labels/class. We use the same hyperparameters for these experiments as those used for homophilous graphs (listed in Table 8).

Table 12: **Results of training GCN with different label rates on heterophilous graphs.** Results are averaged across 100 runs with random train-test splits (subscripts indicate standard deviation).

| Dataset | Labels/Class | GCN | Self-train-GCN | FASTRAIN-GCN |
|---|---|---|---|---|
| Chameleon | 1 | $22.7_{4.9}$ | $26.2_{4.9}$ | $\mathbf{28.8}_{4.5}$ |
| | 2 | $28.4_{3.8}$ | $31.9_{4.5}$ | $\mathbf{35.6}_{4.4}$ |
| | 4 | $33.8_{4.1}$ | $35.1_{3.6}$ | $\mathbf{38.4}_{3.9}$ |
| | 8 | $35.0_{3.2}$ | $36.9_{4.1}$ | $\mathbf{39.3}_{3.7}$ |
| | 16 | $40.5_{2.9}$ | $41.4_{2.2}$ | $\mathbf{42.8}_{3.1}$ |
| Texas | 1 | $28.3_{3.6}$ | $32.4_{4.1}$ | $\mathbf{35.3}_{3.9}$ |
| | 2 | $38.2_{4.8}$ | $41.4_{4.4}$ | $\mathbf{44.7}_{4.7}$ |
| | 4 | $45.5_{2.5}$ | $47.2_{3.1}$ | $\mathbf{49.4}_{2.8}$ |
| | 8 | $50.6_{1.7}$ | $51.5_{1.4}$ | $\mathbf{52.7}_{1.5}$ |
| Wisconsin | 1 | $31.7_{3.4}$ | $33.3_{3.2}$ | $\mathbf{36.4}_{3.8}$ |
| | 2 | $36.0_{4.4}$ | $37.9_{3.8}$ | $\mathbf{40.2}_{4.2}$ |
| | 4 | $44.8_{3.3}$ | $46.4_{2.4}$ | $\mathbf{48.9}_{3.1}$ |
| | 8 | $49.2_{1.3}$ | $50.6_{1.8}$ | $\mathbf{51.7}_{1.9}$ |
| Cornell | 1 | $33.5_{4.4}$ | $36.4_{3.9}$ | $\mathbf{38.7}_{4.4}$ |
| | 2 | $40.1_{4.6}$ | $43.0_{4.0}$ | $\mathbf{45.9}_{3.5}$ |
| | 4 | $47.8_{3.1}$ | $49.1_{4.2}$ | $\mathbf{52.2}_{3.6}$ |
| | 8 | $53.9_{2.5}$ | $54.8_{2.1}$ | $\mathbf{56.1}_{2.1}$ |

We present results on the Chameleon, Texas, Wisconsin and Cornell datasets with different label rates in Table 12. While SPF, DR and DS are used without any modifications, PGP is modified to prune intra-class edges instead of inter-class edges for heterophilous graphs. If we have no prior knowledge about whether the graph is homophilous or heterophilous, we can decipher this from the training set (containing nodes with actual labels and pseudolabels) at the end of the first stage of self-training. The homophily ratio (ratio of edges connecting nodes of the same class to the total number of edges) of edges connecting nodes in the training set indicates whether the graph is homophilious or heterophilious. For instance, Cora (a homophilous graph) has an average homophily ratio of 0.83 among nodes in the training set after the first stage of self-training, while Chameleon (a heterophilous graph) has a homophily ratio of 0.2. We

find that FASTRAIN-GNN consistently outperforms conventional self-training under different label rates on heterophilous graphs also. In addition to producing models that are more accurate, FASTRAIN-GNN also accelerates the self-training process (Table 13).

Table 13: **Accuracy and efficiency gains from the different FASTRAIN-GNN optimizations on Chameleon.** Speedup is computed over self-training of the more accurate among 2/3-layer models, and averaged across different train/test splits and label rates. +Intra-class edges are pruned in PGP instead of inter-class edges.

| Labels/Class | Self-Training-GCN (2-layer/ 3-layer GCN) | with SPF and DR (2-layer/ 3-layer GCN) | with SPF, DR and DS | with SPF, DR, DS and PGP+ |
|---|---|---|---|---|
| 1 | $23.4_{3.6}$ / $26.2_{4.9}$ | $24.0_{3.2}$ / $26.8_{3.8}$ | $27.7_{3.9}$ | $28.8_{4.5}$ |
| 2 | $28.7_{2.9}$ / $31.9_{4.5}$ | $30.0_{3.3}$ / $33.2_{4.1}$ | $34.5_{4.6}$ | $35.6_{4.4}$ |
| 4 | $33.9_{3.1}$ / $35.1_{3.6}$ | $35.0_{3.0}$ / $36.2_{2.9}$ | $37.1_{4.1}$ | $38.4_{3.9}$ |
| 8 | $36.9_{4.1}$ / $36.1_{3.9}$ | $38.0_{3.8}$ / $37.5_{3.6}$ | $38.6_{3.5}$ | $39.3_{3.7}$ |
| 16 | $41.4_{2.2}$ / $38.7_{1.9}$ | $41.9_{2.4}$ / $39.9_{1.1}$ | $41.9_{2.4}$ | $42.8_{3.1}$ |
| **Average Speedup** | 1X | 1.2X | 1.7X | 1.9X |

