# OpenReview forum: "FASTRAIN-GNN: Fast and Accurate Self-Training for Graph Neural Networks"
_TMLR — Accepted by TMLR_

### Review · Reviewer_PMSV · 2023-01-23

**Summary Of Contributions:**

This paper proposed FASTRAIN-GNN, which is an improved framework for the self-training of GNNs in terms of learning performance and speed, mainly for transductive node classification tasks. It consists of four major parts:  Sampling-based Pseudolabel Filtering,  Dynamic Regularization, Dynamic Sizing, and Progressive Graph Pruning. The motivation for these parts is from the intuition and experiments on empirical graph datasets about the quality of pseudo-labels,  graph sampling/pruning, layer effects (over-smoothing), and homophily of the graph.

**Audience:**

Yes

**Broader Impact Concerns:**

I have no broader impact concerns.

**Claims And Evidence:**

Yes

**Requested Changes:**

**Major**:
1. In Section 3, it is better to use some concise words to describe the method instead of such long paragraphs. Some mathematical equations or the format of algorithms are helpful. Actually, I can feel that in this part, the authors focus on introducing more about intuition and the understanding of the method. Therefore, a complete algorithm for DR and DS is expected to know the complete architecture and idea of the technique (just like Algorithms 1 and 2).
2. Some experimental results are quite close between curves, e.g., Figure 3, 5, 8; Table 2, 3. The setup of random seeds and the error bar might be necessary for these figures and tables.
3. I recommend adding some introduction about the lottery ticket hypothesis because the methods of SPF and PGP are related. I also recommend adding a discussion of related works in Section 2. Some references include:
(a) https://proceedings.neurips.cc/paper/2017/file/5dd9db5e033da9c6fb5ba83c7a7ebea9-Paper.pdf,
http://proceedings.mlr.press/v139/chen21p/chen21p.pdf,
https://arxiv.org/pdf/1801.10247.pdf%EF%BC%89 (The authors have cited them, but introducing it in Section 2 is also appropriate.)
(b) https://proceedings.neurips.cc/paper/2020/file/d714d2c5a796d5814c565d78dd16188d-Paper.pdf,
https://proceedings.mlr.press/v162/li22u/li22u.pdf,   https://openreview.net/forum?id=4UldFtZ_CVF. (These are theoretical understanding of efficient graph learning with sampling, from training and generalization.)
4. From the testing datasets and the intuition of the PGP method, I feel the proposed method is mainly for homophilous graphs. It is unsure how the proposed tool works on heterphilous graphs. Will you consider adding virtual edges between nodes in any case? Discussions about limitations and future works are needed.

**Minor**:
1. On page 7, in the second to last line in the paragraph above Table 2, it should be "Table 1" instead of "Table 4", right?


**Strengths And Weaknesses:**

**Strength**:
1. The intuition in the paper makes sense to me, and it is also verified by experiments.
2. The empirical experiments, especially the comparison with existing methods, show the advantages of the proposed technique in terms of performance and training speed.
3. The writing is clear. It is easy to follow.

**Weakness**:
1. The writing is not concise enough.
2. The experiments need improvement in terms of presentation.
3. Some necessary motivation and introduction might be missing in the introduction section.
4. Some discussion about limitations and future directions is missing.

---

> ### Author Response · Authors · 2023-02-26
> **Response to Reviewer PMSV**
>
> We thank the reviewer for the thoughtful review. We address the main concerns below.
>
> **Re: A complete algorithm for DR and DS is expected to know the complete architecture and idea of the technique (just like Algorithms 1 and 2)**
>
> We have added Algorithm 2 to “The FASTRAIN-GNN Framework” section to describe the implementation of DS and DR. Algorithms 1, 2 and 3 together now describe the complete procedure for all the FASTRAIN-GNN optimizations in the updated paper. We have also added a “Hyperparameter Tuning” section in Appendix D that lists the exact hyperparameter values used in all of our experiments, while also providing additional insights from our experience on tuning these hyperparameters.
>
> **Re: Some experimental results are quite close between curves, e.g., Figure 3, 5, 8; Table 2, 3. The setup of random seeds and the error bar might be necessary for these figures and tables**
>
> We thank the reviewer for pointing out that the differences between curves were not clear. We have added error bars to all the tables and figures whose results are close to each other. We have also redone figures 3, 5, 8 and 9 to better depict the main ideas.
>
> **Re: I recommend adding some introduction about the lottery ticket hypothesis because the methods of SPF and PGP are related. I also recommend adding a discussion of related works in Section 2**
>
> We’ve added some details about the lottery ticket hypothesis and sampling methods in both the intro (last paragraph) and the related works sections. We also thank the reviewer for pointing out the missing references. We’ve incorporated them into the updated paper.
>
> **Re: From the testing datasets and the intuition of the PGP method, I feel the proposed method is mainly for homophilous graphs. It is unsure how the proposed tool works on heterphilous graphs. Will you consider adding virtual edges between nodes in any case?**
>
> PGP only for homophilous graphs – The reviewer is indeed correct that pruning inter-class edges through PGP is applicable only to homophilous graphs, where connected nodes are expected to have similar labels. In the case of heterophilous graphs, connected nodes have dissimilar labels, and hence, pruning inter-class edges is highly undesirable. In graphs that rely on the heterophily assumption, we demonstrate that a slightly modified version of PGP -- pruning intra-class edges instead of inter-class edges -- leads to accuracy gains by reducing the effect of over-smoothing in Appendix 6 in the updated paper. In particular, we have added results on 4 popular heterophilous graph datasets (Chameleon, Texas, Wisconsin and Cornell) in Tables 12 and 13. We’ve discussed this in greater detail in the newly added “Discussion” section in the updated paper. The  “Discussion” section also lists out the limitations of FASTRAIN-GNN, and outlines directions for future work.
>
> Adding virtual edges – In FASTRAIN-GNN, we only consider optimizations that simultaneously improve both accuracy and computational efficiency. While adding virtual edges is likely to improve accuracy, it also hurts efficiency due to neighborhood expansion from virtual edges.
>
> We hope we have addressed the main concerns. If not, please let us know and we will be happy to address them in subsequent responses.

---

> > ### Comment · Reviewer_PMSV · 2023-02-28
> > **Thank the authors for the responses and revisions**
> >
> > Thank the authors for the responses and revisions. Most of my concerns have been addressed.

---

### Review · Reviewer_XEHk · 2023-01-24

**Summary Of Contributions:**

This paper introduces a framework for fast and accurate self-training of GNNs using unlabeled data. Specifically, first, it proposes the sampling-based Pseudo-label Filtering approach to actively select pseudo-labeled data with a high confidence score. Second, it proposes the Dynamic Sizing and Dynamic Regularization to reduce the oversmoothing problem raised by multi-layer GNN and residual error in pseudo-labels.  Third, it proposes the Progressive Graph Pruning approach to address the oversmoothing problem raised by the inter-class edges.

**Audience:**

Yes

**Broader Impact Concerns:**

I did not see any broader impact concerns in this paper.

**Claims And Evidence:**

Yes

**Requested Changes:**

I hope the author provides either theoretical computation or numerical experiments to justify my concerns in the weakness.

**Strengths And Weaknesses:**

Strengths:

1. The intuitions for the algorithm are well-presented.

2. The experiments results justify the efficiency and accuracy of the proposed self-training approach for GNNs.

Weakness:

1. The training time of GNNs increases dramatically as the training samples increase. Could the author report the improved test accuracy (over training GNN using labeled data only) against the enlarged training time (over training GNN using labeled data only) by varying the number of unlabeled data used in the experiments? Therefore, we could have a better idea for the efficiency of using self-training in GNN.

2. ResGNN can overcome the oversmoothing issue caused by the layers. Could the author discuss the possibility of using ResGNN in your framework?

3. Although the intuitions are well-presented, the implementation of the algorithm is not clear to the readers. Some sampling questions, (1) what sampling rate should we choose in practice? Can we keep the same sampling rate if we change the dataset? (2) when decreasing the layer number and sampling rate as the training samples increase, how should we decrease them based on the number of training samples? Therefore, I hope the authors can explore more about the proposed algorithms, for example, (2.1) what is the benefits of selecting a large (or small) sampling rate in SPF? when should we choose a relatively large (or small) sampling rate in SPF? Besides sampling, is there any ways in existing works to test the robustness of GNNs, and how do you compare your methods with theirs? (2.2) how the threshold in DS is derived? how should we select the threshold given different datasets? would it be better to set multiple sets of the threshold and corresponding GNN model in the algorithms?  (2.3) how do we quickly detect the inter-class edges? What is the relationship between the computational times in detecting inter-class edges and computing self-training using the sampled nodes?

---

> ### Author Response · Authors · 2023-02-26
> **Response to Reviewer XEHk (1/2)**
>
> We thank the reviewer for the insightful review. We address the main concerns below.
>
> **Re: Could the author report the improved test accuracy (over training GNN using labeled data only) against the enlarged training time (over training GNN using labeled data only) by varying the number of unlabeled data used in the experiments?**
>
> We have added this graph (Figure 11) in Appendix A to discuss the computational demands of self-training in the updated paper. In summary, self-training can lead to substantial accuracy gains, but it comes at a large computational cost, thereby motivating the need for efficient self-training methods.  We also analyze the impact of the number of stages of self-training on both accuracy and wall-clock training time. We find that when the number of stages is increased beyond a certain number, the accuracy gain saturates, while the wall-clock training time continues to increase.
>
> **Re: ResGNN can overcome the oversmoothing issue caused by the layers. Could the author discuss the possibility of using ResGNN in your framework?**
>
> Our framework is designed to work with any GNN variant. However, the choice of architecture for deep_gnn and shallow_gnn depends on the GNN variant, since different models require different numbers of layers for best accuracy. While ResGCN does not suffer from oversmoothing to the same extent as GCN, it is still an issue in ResGCN. For instance, in DropEdge (https://github.com/DropEdge/DropEdge), the authors observe that 4-layer ResGCN models typically outperform deeper models in the semi-supervised setting, indicating that oversmoothing is still an issue.
> We also found a different model named Res-GNN (https://arxiv.org/pdf/2101.08543.pdf). If the reviewer was referring to this model instead, please let us know.
>
>
> **Re: Although the intuitions are well-presented, the implementation of the algorithm is not clear to the readers**
>
> We have added Algorithm 2 to “The FASTRAIN-GNN Framework” section. Algorithms 1, 2 and 3 together now describe the complete procedure for all the FASTRAIN-GNN optimizations in the updated paper. We have also added a “Hyperparameter Tuning” section in Appendix D that lists the exact hyperparameter values used in all of our experiments, while also providing additional insights from our experience on tuning these hyperparameters.
>
> **Re: What sampling rate should we choose in practice? Can we keep the same sampling rate if we change the dataset?**
>
> The choice of sampling_rate_init (sampling rate in the first stage of self-training, when only golden training labels are used) has been extensively studied in prior works, since sampling is a popular regularizer for training GNNs. We find that sampling_rate_init=0.9 works well across the different datasets, label rates and GNN architectures used in our experiments. Similarly, we find that sampling_rate_final=0.8 works well in all our experiments by finding a good balance between preventing overfitting to noisy labels and ensuring convergence during training. We explain these choices in greater detail in Appendix D.2 in the updated paper.
>
> **Re: When decreasing the layer number and sampling rate as the training samples increase, how should we decrease them based on the number of training samples?**
>
> We have added Algorithm 2 to “The FASTRAIN-GNN Framework” section that denotes the exact procedure for performing DS and DR. The hyperparameter values used in our experiments are listed in Table 8 in Appendix D.
>
> **Re: What is the benefits of selecting a large (or small) sampling rate in SPF? When should we choose a relatively large (or small) sampling rate in SPF?**
>
> When spf _sampling_rate is too large (>=0.95), we find that there is insufficient perturbation to the graph structure to distinguish between robust and non-robust predictions. As a result, the predictions of the vast majority of candidate nodes remain constant across sampled configurations, and SPF becomes ineffective. On the other hand, when spf _sampling_rate is too small (<=0.8), the number of nodes with consistent predictions decreases drastically. The primary reason for the high effectiveness of self-training for few-shot transductive node classification is the increased number of nodes taking part in the training process
> through training set augmentation (either because they are added to the training set, or because they are neighbors of nodes added to the training set). As a result, the decreased coverage of nodes taking part in training with a small spf _sampling_rate outweighs the benefits of more accurate training labels. In our experiments, we find that spf _sampling_rate = 0.9 provides the best trade-off between coverage of nodes seen during training and the accuracy of training labels. We explain this choice in greater detail in Appendix D.1 in the updated paper.

---

> > ### Author Response · Authors · 2023-02-26
> > **Response to Reviewer XEHk (2/2)**
> >
> > **Re: Besides sampling, is there any ways in existing works to test the robustness of GNNs, and how do you compare your methods with theirs?**
> >
> > Prior efforts on testing the correctness of GNN predictions involve training a second model along with the GNN, and ensuring that the GNN’s predictions match those of the second model. The “union” and “intersection” methods introduced by Li et. al (2018) are notable examples of this approach. However, these approaches incur significant computational costs, since two models need to be trained instead of one. In contrast, SPF can quickly filter out nodes that are likely to be incorrectly classified, thereby adding minimal overheads to the self-training process (see Appendix E).
> >
> > **Re: How the threshold in DS is derived? How should we select the threshold given different datasets?**
> >
> > The DS_Threshold is used to indicate the point at which oversmoothing outweighs the benefits of increased message passing coverage in deeper GNNs. We empirically find that when the message passing coverage from training nodes is >75% (more than three-fourths of all nodes in the graph take part in training, either because they are added to the training set, or they are neighbors of nodes in the training set) in shallow GNNs, they outperform deep GNNs. We find that holds across the different datasets, label rates and GNN architectures used in our experiments. We also find that this holds irrespective of the choice of initial training nodes. Since all datasets used in our experiments exhibit similar (regular) connectivity patterns, we find that when approximately 2.5% of all nodes in the graph are used for training, the 2-hop message passing coverage becomes greater than 75%. We discuss this in greater detail in Appendix D.3 in the updated paper.
> >
> > **Re: How do we quickly detect the inter-class edges? What is the relationship between the computational times in detecting inter-class edges and computing self-training using the sampled nodes?**
> >
> > We have added Appendix E in the updated paper to discuss the overheads introduced by the different FASTRAIN-GNN optimizations. At the end of each stage of self-training, PGP analyzes the labeled neighbors of nodes that were added to the training set in that stage to identify inter-class edges. Since 1-hop neighbors of labeled nodes can be quickly identified from the adjacency matrix, PGP only adds small overheads, typically accounting for <2% of the total runtime. On the other hand, training the GNNs in different stages accounts for >80% of the total runtime.
> >
> > We hope we have addressed the main concerns. If not, please let us know and we will be happy to address them in subsequent responses.

---

> ### Author Response · Authors · 2023-03-14
> **Follow-up**
>
> We appreciate your insightful comments about our work. We were wondering if our rebuttal addressed your concerns. If not, we will be happy to address them in subsequent responses and revisions.

---

### Review · Reviewer_1Nc3 · 2023-02-18

**Summary Of Contributions:**

This paper focused on self-training GNNs for few-shot transductive node classification. The proposed FASTRAIN-GNN introduced four major optimizations in each stage of self-training:
(1)	Sampling based Pseudolabel Filtering to generate more reliable pseudo labels.
(2,3) Dynamic Sizing and Dynamic Regularization to overcome the layer effect and reduce the impact of incorrect training labels.
(4) Progressive Graph Pruning to remove inter-class edges and improve information-to-noise ratio during message passing.
The approach is claimed to accelerate the self-training as well as to improve the accuracy.


**Audience:**

Yes

**Broader Impact Concerns:**

The paper may bring some ideas to improve self-training of GNNs, but it does not have enough theoretic or empirical justification.

**Claims And Evidence:**

No

**Requested Changes:**

1. adding more theoretic explanation why the proposed module is useful

2. experiments on different types of datasets (especially heterophily graphs) and bigger datasets


**Strengths And Weaknesses:**

Strengths:

* The paper is generally written clearly and easy to follow.

* Some of the proposed modules seem new and effective, e.g. the sampling based pseudolabel filtering.

* I like the experimental results comparing different label rates.

Weaknesses:

* Although the paper proposed four optimization procedures for self-training, none of them is attractive enough. In other words, the paper lacks a striking point, which makes it more like an engineering pipeline.

* No clear explanation of the motivation and no theoretic justification of the proposed method. SPF is seemingly interesting, but it lacks a theoretic justification; similarly, DR has the same issue. For example, the authors claim, “we find that sampling is an effective regularizer”, but why? It does not have a detailed discussion. Without detailed explanation of the motivation and theoretic advantages, the proposed modules will only be regarded as heuristic tricks and novelty will look trivial.

* One big issue of the experiments are the datasets. One of the focuses of this paper is to accelerate the self-training, but all the datasets are relatively small in the graph learning community; and all of them actually have enough labels. To demonstrate the proposed method is practically useful, readers may be more convinced if they see a practical scenario with bigger data size and less labels. Moreover, all the datasets in this paper are citation datasets but we do not know whether the method also work for other types of graph datasets (especially heterophily graphs). That makes the generalizability of the method also questionable.

---

> ### Author Response · Authors · 2023-02-26
> **Response to Reviewer 1Nc3**
>
> We thank the reviewer for the feedback. We address the main concerns below.
>
> **Re: Experiments on heterophilous graphs**
>
> We have added results on 4 popular heterophilous graph datasets (Chameleon, Texas, Wisconsin and Cornell) in Tables 12 and 13 (Appendix F) in the updated paper. While SPF, DR and DS remain unchanged, we use a slightly modified version of PGP for heterophilous graphs -- pruning intra-class edges instead of inter-class edges. We’ve discussed this in greater detail in the newly added “Discussion” section in the updated paper. We find that FASTRAIN-GNN consistently outperforms conventional self-training in terms of accuracy and computational efficiency on heterophilous graphs also.
>
> **Re: Experiments on bigger dataset with less labels**
>
> We note that all prominent prior works on self-training (https://arxiv.org/abs/1801.07606, https://arxiv.org/abs/1902.11038, https://arxiv.org/abs/2109.14285 etc.) present results only on Cora, Citeseer and Pubmed for different label rates. We also experiment on these datasets to provide a fair comparison to prior work. We do not understand the reviewer’s comment on these datasets already having “enough labels”. All samples in any dataset will have labels, and the train-test ratio is reduced to show results in the few-shot setting. If a dataset does not have sufficient labels, it will be impossible to test the accuracy. However, if the reviewer would like to see results on specific datasets, we are happy to do the experiments.
>
> **Re: Theoretical justification for the proposed methods**
>
> We respectfully disagree with the reviewer’s characterization of empirical methods that are not backed by theoretical guarantees as “not attractive enough” or “heuristic tricks”. We believe that our framework, which combines novel empirically-driven methods to advance the state-of-the-art in few-shot transductive node classification by up to 4.4%, while also simultaneously reducing training time by up to 2.1X, presents a valuable empirical contribution. While we do not provide theoretical guarantees for our methods, we explain the intuitions in detail and provide empirical validations for all the proposed methods (explicitly called out as strengths of the paper by the other two reviewers). To the best of our understanding, contributions with an empirical focus and basis are within the scope of TMLR, especially since we do not claim to make any theoretical contributions in the paper. If the reviewer feels that any particular claim or method is not empirically justified, please let us know and we will address it.
>
> **Re: “We find that sampling is an effective regularizer”, but why?**
>
> We thank the reviewer for pointing out this missing motivation, and we have added it to the updated paper. Increased regularization is a popular method for preventing overfitting to noisy labels (for example, see https://proceedings.neurips.cc/paper/2020/file/ea89621bee7c88b2c5be6681c8ef4906-Paper.pdf), and sampling is the prominent regularization method for GNNs. Since a random subset of edges are pruned in each training iteration, sampling can prevent memorization of noisy labels. This can be empirically seen from the fact that smaller sampling rates lead to more accurate models in the presence of noisy labels (Figure 3).
>
> We hope we have addressed the main concerns. If not, please let us know and we will be happy to address them in subsequent responses.

---

> > ### Comment · Reviewer_1Nc3 · 2023-03-29
> > **Response to authors**
> >
> > Thank you for the clarification and new experiments.
> > 1. My concerns about performance on heterophilous graphs or bigger dataset have been mostly solved. However, in practice we do not know whether a graph is homophilious or heterophilious, how do we decide to delete the inter-class or intra-class edges? Also as the authors mentioned, it is not easily modified when both types of edges are prominent. So, the PGP seems not very practical or valuable. But it is good that the authors discuss the different cases and limitations.
> > 2. As to the explanation of the why sampling is an effective regularizer, the new content looks good. I suggest the authors to add one sentence in the main paper to indicate that there are more detailed explanation in Appendix.
> > 3. I agree that theoretic justification is not a must if the motivation is insightful and explained well. However, my impression that the proposed method looks like an assembled tricks is not changed. It is just not very attractive, but I admit its usefulness to the self-learning community, and I do not object if others would accept it.

---

> > > ### Author Response · Authors · 2023-03-30
> > > **Reply**
> > >
> > > **Re: In practice we do not know whether a graph is homophilious or heterophilious, how do we decide to delete the inter-class or intra-class edges?**
> > >
> > > If we have no prior knowledge about the graph, we can decipher this from the training set (containing nodes with actual labels and pseudolabels) at the end of the first stage of self-training. The homophily ratio (ratio of edges connecting nodes of the same class to the total number of edges) of edges connecting nodes in the training set indicates whether the graph is homophilious or heterophilious. For instance, we find that Cora (a homophilous graph) has an average homophily ratio of 0.83 among nodes in the training set after the first stage of self-training, while Chameleon (a heterophilous graph) has a homophily ratio of 0.2. We have added this clarification in Appendix F in page 21, colored in orange.
> > >
> > > We have added a sentence in the main paper indicating that more details are available in the appendix (section 3 in page 5, colored in orange). We thank the reviewer for engaging in discussions that help further improve our paper. We are also happy to provide additional clarifications.

---

> ### Author Response · Authors · 2023-03-14
> **Follow-up**
>
> We appreciate your insightful comments about our work. We were wondering if our rebuttal addressed your concerns. If not, we will be happy to address them in subsequent responses and revisions.

---

### Decision · Action_Editors · 2023-03-29

**Recommendation:** Accept with minor revision

**Comment:**

This paper studied different optimization methods for improving graph neural networks in self-training settings. In the first round, all reviewers made very constructive comments about the submission. The authors did a good job revising and improving the paper, and most of the major concerns have been addressed.

However, there are several post-rebuttal suggestions on adding additional discussions to further clarify the claims (especially from Reviewer 1Nc3). I deemed the changes as necessary and therefore recommend "accept with minor revision". I ask the authors to prepare a response and post it on openreview to explain how they incorporate the reviewers' comments and suggestions in the final version.



**Audience:**

Suited for readers interested in graph learning

**Claims And Evidence:**

This paper studied different optimization methods for improving graph neural networks in self-training settings. The reviewers have made several suggestions to improve the initial version, and the authors have greatly improved the paper in the revision. The experiments are sufficient to support their claims.

---

> ### Author Response · Authors · 2023-04-17
> **Camera-ready Paper Upload**
>
> We thank the AE and reviewers for helping improve our paper. We have now uploaded the camera-ready paper. We have also responded to all reviewer comments that are visible to us, and incorporated the necessary changes in our paper.